ecology/health and disease and epidemiology

citizen science, venomous snakebite, misidentification, biodiversity, item response theory, online challenge

**Author for correspondence:**
A. M. Durso
e-mail: amdurso@gmail.com

# Crowdsourcing snake identification with online communities of professional herpetologists and avocational snake enthusiasts

A. M. Durso[1,2], I. Bolon[1], A. R. Kleinhesselink[3],
M. R. Mondardini[4], J. L. Fernandez-Marquez[5],
F. Gutsche-Jones[4], C. Gwilliams[4], M. Tanner[4],
C. E. Smith[6], W. Wüster[7], F. Grey[5]
and R. Ruiz de Castañeda[1]

[1]Institute of Global Health, Department of Community Health and Medicine, Faculty of Medicine, University of Geneva, Geneva, Switzerland
[2]Department of Biological Sciences, Florida Gulf Coast University, Ft. Myers, FL, USA
[3]Department of Ecology and Evolutionary Biology, University of California Los Angeles, Los Angeles, CA, USA
[4]Citizen Science Center Zürich, ETHZ and University of Zürich, Zurich, Switzerland
[5]Citizen Cyberlab, University of Geneva, Geneva, Switzerland
[6]HerpMapper, MN, USA
[7]Bangor University College of Natural Sciences, Bangor, UK

AMD, 0000-0002-3008-7763

Species identification can be challenging for biologists, healthcare practitioners and members of the general public. Snakes are no exception, and the potential medical consequences of venomous snake misidentification can be significant. Here, we collected data on identification of 100 snake species by building a week-long online citizen science challenge which attracted more than 1000 participants from around the world. We show that a large community including both professional herpetologists and skilled avocational snake enthusiasts with the potential to quickly (less than 2 min) and accurately (69–90%; see text) identify snakes is active online around the clock, but that only a small fraction of community members are proficient at identifying snakes to the species level, even when provided with the snake's geographical origin. Nevertheless, participants showed great enthusiasm and engagement, and our study provides evidence that innovative citizen science/crowdsourcing approaches can play

significant roles in training and building capacity. Although identification by an expert familiar with the local snake fauna will always be the gold standard, we suggest that healthcare workers, clinicians, epidemiologists and other parties interested in snakebite could become more connected to these communities, and that professional herpetologists and skilled avocational snake enthusiasts could organize ways to help connect medical professionals to crowdsourcing platforms. Involving skilled avocational snake enthusiasts in decision making could build the capacity of healthcare workers to identify snakes more quickly, specifically and accurately, and ultimately improve snakebite treatment data and outcomes.

# 1. Introduction

Species identification is challenging due to the complexity and vastness of biodiversity. Species misidentification introduces both false-positive and false-negative errors into biological datasets spanning taxa from mammals [1–3] and birds [4] to fishes [5], invertebrates and plants [6]. Species identification skills can only be obtained through intensive training and experience, and the number of taxonomists and other trained experts is declining [7]. Even experts may sometimes be responsible for species misidentification [8,9], such that species misidentification has been explicitly incorporated as a source of error into some wildlife monitoring programmes [10,11].

Snake identification can be especially fraught [12,13] as many snakes resemble and mimic other species of snakes to which they may or may not be closely related [14]. The number of described species of snakes is growing rapidly, now more than 3800 species globally with approximately 30 new species described every year since the turn of the century, ca 20% of which are potentially dangerous to humans [15]. This renaissance in new species discovery has been brought about through globalization, digitization of museum collections, increased exploration of previously neglected areas, and the integrated use of morphological and molecular techniques for species delimitation [16]. Snake species identification is also complicated by geographical and ontogenetic variation in morphology and colour pattern, and the fact that many people are frightened of snakes and have not had much exposure to them.

Rapid, accurate identification of snakes to the species level is a critical need in snakebite epidemiology and clinical management of snakebite cases [17,18]. Snakebite envenoming (SBE) is a life-threatening, neglected tropical disease, with an underestimated but substantial health burden (estimated 81 000– 138 000 fatalities and approx. 400 000 victims surviving with disabilities every year worldwide; [19]). Snake bites are emergencies that require rapid, precise treatment. Despite recent progress in pan-specific antivenom development [20] all available antivenoms are monovalent (species-specific) or polyvalent (covering a limited number of species, normally fewer than 10). Thus, healthcare providers often must determine whether to treat the patient with specific antivenom and/or other life-saving interventions (e.g. ventilator support). When the biting species is unknown, healthcare providers must rely on matching clinical syndromes of envenoming with snake species and specific antivenoms [18,21]. Overlap in clinical features caused by the venoms of different snake species [22,23] is a significant limitation, and there is often not sufficient time to consult unwieldy dichotomous keys, deploy molecular techniques for identification from DNA or proteins left at the bite site, or wait for consultation with a limited number of experts who may live half a world away. A recent review found that snakes were identified at the species or genus level in only 53% of snakebite cases, and found 106 misidentifications that led to inadequate victim management [18]. Identifying the biting snake can support healthcare providers in their decision and help them anticipate symptoms, as well as provide essential data for retrospective studies aiming to establish species-syndrome correlations and understand the epidemiology of snake bites.

Despite some evidence that some lay people can correctly classify snakes as being venomous or non-venomous [24], species-level identification by members of the general public [25–27], university science and medical students [28], and patients and healthcare providers [13,18,29] is much less accurate. The species-specific nature of many antivenoms and the potentially dire consequences of antivenom misuse (e.g. [30–33]) have led to the belief that biting snakes should be captured and transported to the treatment centre for definitive identification at more specific taxonomic levels [18]. However, healthcare providers are not trained to identify snakes and qualified snake experts are not always available. This practice has several drawbacks, including risk of a second bite (including from dead snakes; [34,35]) and delaying transport to medical care, as well as being undesirable from the perspective of snake conservation [36]. Identification of snakes from

photographs is likely to be quicker and safer [18]. The increasing proliferation of smart phones with cameras even in many poor tropical regions provides new opportunities to ensure the identification of biting snakes, either rapidly in an ongoing situation, or for later, for instance for the retrospective assessment of antivenom effectiveness.

Improving snakebite treatment outcomes is an ambitious but achievable goal [19]. In order to do so, we need to assess the efficacy of antivenoms by looking at past medical records. However, because the taxonomic identities of snakes are unavailable, incomplete or vague in many studies on snakebite [37], even in high-income countries [38], assessing the efficacy and paraspecificity of antivenoms retrospectively is difficult to impossible, making development of species-specific evidence-based treatment guidelines for the management of snakebite extremely challenging [13,19]. This includes identification of non-venomous snakes, which are much more diverse and abundant than medically significant venomous species (MIVS) in most parts of the world. Accurate and objective identification may be especially important when bites from non-medically significant snakes with mild to moderate local or transient effects [39] are confused with those from dangerous snakes [40,41] and no antivenom is required. Even bites from other animals that superficially resemble snakes (e.g. amphisbaenians) may be received with alarm by clinicians if patients or their families are hysterical [33].

Although snakebite can be dangerous, snakes are also ecologically important and fascinating, and there are worldwide communities of both professional herpetologists and avocational snake enthusiasts who study and enjoy learning to identify snakes. Online communities focused on snakes include forums (e.g. FieldHerpForum.com), citizen science projects such as HerpMapper.org and iNaturalist.org (e.g. the medically important venomous snakes (MIVS) project; [42]), followers of certain hashtags on Twitter (e.g. #notacopperhead), and Facebook groups (approx. 50 groups focused on snake identification, notably 'Snake Identification' with greater than 200 000 members [43] as well as groups for sharing natural history observations about snakes [44]). These communities involve a relatively small number of professionals (e.g. scientific researchers, museum curators, government agency personnel) as well as many skilled amateurs, hobbyists, herpetoculturists and other avocational snake enthusiasts with some degree of training or experience identifying snakes [17]. A larger number of more ephemeral community members with essentially no knowledge or experience make individual posts seeking identification, which are then identified by the more experienced members. The more experienced community members, who have a lot of knowledge and experience but normally not much formal training (although a still smaller number may have a great deal of formal training or even be working as herpetologists), play a key role in these communities by providing identifications and information or tagging other members in their network who may be able to help [45,46].

Typically, a photo of a snake is posted in an online community without a conclusive identification but with the rough geographical location (e.g. country, state/province), and the experienced members of the group discuss and provide an identification, usually within a few minutes [43]. All members learn through the day-to-day activities of the group, and some members who began as complete novices have become advanced and participate in providing identifications and managing the group. These communities are very active (sometimes more than 400 identifications per day), and many of the more experienced members are extremely enthusiastic and passionate about snake biodiversity, ecology and conservation. Data from these communities have begun to be used to inform snake ecology [47,48]. Community members are also already involved, informally, in occasional consultations with snakebite victims (both human and domestic animal) or their healthcare providers, who take pictures of dead or living snakes and are seeking expertise in snake identification [43]. There are even specific snakebite support groups (electronic supplementary material, table S1). Using citizen science to crowdsource animal identification has had great success for other taxa with large and highly engaged non-professional communities (e.g. birds; [49,50]; butterflies; [51]).

The use of crowdsourcing in healthcare to support both medical professionals (e.g. CrowdMed, HealthTap) and patients (e.g. PatientsLikeMe) in the diagnosis of complex health conditions is growing [52,53]. Crowdsourcing may be especially helpful in the context of snakebite, where the number of trained health professionals is limited and affected populations are severely neglected. Healthcare providers managing snakebite in the field already share photos of biting snakes with their medical colleagues and with herpetologists they may know and trust, at a small scale but increasingly beyond their immediate networks. Yet, crowdsourcing a diagnosis (here, a snake identification), whether used for clinical management or research in snakebite epidemiology, raises important legal and ethical risks that must be carefully addressed [54]. Healthcare providers should be ultimately responsible for any clinical decision, and an ideal decision support system could generate a consensual output by considering all responses from the community and weighing the level of

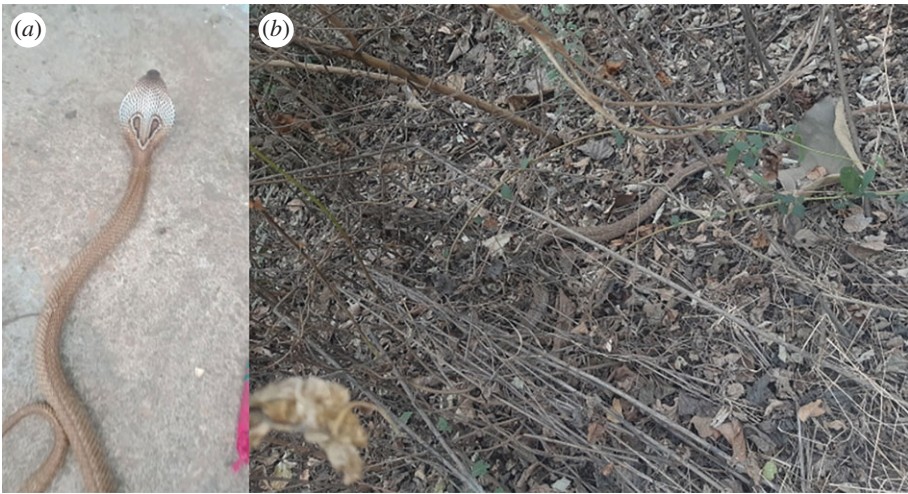

**Figure 1.** High-quality (*a*) and low-quality (*b*) images of *Naja naja* (Indian cobra), taken from Indian Snakebite Initiative database, courtesy of J. Louies (indiansnakes.org). The snake in the high-quality image (*a*) fills most of the frame, is against a uniform background, almost the entire body is visible and identifying features are clearly visible. The snake in the low-quality image (*b*) fills only a small part of the frame, is against a complex background, only part of its body is visible, and identifying features such as the hood and its markings are obscured.

expertise of each contributor (e.g. based on snake identification challenges such as the one we present here, community reputation, or an initial questionnaire on snake identification).

The objectives of our study were to (i) assess the capacity of online mixed professional-avocational snake enthusiast communities to quickly and accurately identify a diversity of snake species from photos, (ii) to understand the relative importance of the main factors that affect this process (geography, photo quality, snake taxonomy, variation among participants), and (iii) assess the potential role of citizen science and crowdsourcing in supporting snakebite epidemiology and management.

## 2. Methods

We used the citizen science platform developed by the Citizen Science Center Zurich and Citizen Cyberlab (based on the Pybossa open framework; [55]) to host a snake identification online challenge (snakes.citizenscience.ch) which participants could play as a game [56]. We selected 10 photos each of 100 widespread snake species from known locations around the world, for a total of 1000 photos, representing 130 countries from all continents and all major taxonomic groups of snakes, including both medically important venomous snakes (MIVS) and non-venomous snakes (electronic supplementary material, table S2). Photos were gathered primarily from online open platforms with a focus on snakes, especially HerpMapper, iNaturalist and Facebook groups on snake identification. A herpetologist (A.M.D.) scored each photo as high-quality or low-quality, based on the focus, lighting, angle and percentage of the snake visible in the frame, and as part of the gamification approach, participants were allowed to select high- or low-quality photos to identify (see figure 1 for examples). We showed photos one at a time to participants, together with the geographical location (continent, country, state/province) of the photo. In addition to the 1000 photos detailed above, photos of 100 additional snake species (one photo each) were presented at random throughout the challenge but not used for analysis, in order to reduce the likelihood that participants would become too familiar with the limited set of correct answers, which would lead to overestimates of accuracy. Using the September 2018 taxonomy of The Reptile Database [15], including relevant taxonomic synonyms, participants could enter or select from a drop-down list a common name (e.g. 'puff adder'), scientific binomen (e.g. '*Bitis arietans*'), genus (e.g. '*Bitis*'), or family name (e.g. 'Viperidae'). Participants were given extra points for correctly identifying low-quality photos. Participants were given unlimited time to suggest an identification for each photo, were told that they were allowed to use resources, and could skip photos if they did not wish to provide an identification (equivalent to saying 'I don't know'). Because they could not choose their global region, all participants had to identify snakes from all regions of the world, regardless of their regional expertise. The maximum number of photos a participant could identify was 1100 (10 photos of each of the 100 core species + 1 each of

100 'distractor' photos) and photos that were skipped reappeared once a participant had identified all other photos. The top score was displayed on the platform to motivate participants to identify many photos. A forum was available for participants to provide photo-specific feedback, and feedback was also collected via social media groups where the challenge was advertised and promoted.

The challenge ran from 25 February to 3 March 2019 and was advertised primarily through email and over social media channels specific to citizen science and snakes. Because we aimed to recruit avocational snake enthusiasts, incentives included a digital badge (an icon that appears on a user's account) for all participants having a HerpMapper account and a herpetological book [57] offered as a prize to the three participants who correctly identified the most photos. All data collected were anonymized by replacing user names and email addresses with unique 36-digit strings. Our study was outside the scope of human subjects institutional review board, as confirmed by the Swiss Human Research Act (HRA) and Cantonal Ethics Committee Geneva (CCER). The research objectives were outlined on the challenge page and in challenge advertisement, and Terms of Use as well as a Privacy Policy are linked at the bottom of all pages of snakes.citizenscience.ch.

## 2.1. Data preparation

We restricted our statistical analysis to only users who filled in their home region information (Africa, Asia, Australia/Oceania, Europe, North America or South America) and who responded to at least 80 photos ($N = 250$). We analysed only survey responses to the 100 core species and excluded responses to the distractor photos ($N = 103\,160$ responses). Prior to analysis we randomly split responses stratified by user into a training (80% of the data) and testing (20% of the data). The testing dataset was held out during model fitting and used to evaluate out-of-sample model predictive ability.

## 2.2. Bayesian item response theory model

To analyse test responses, we used a Bayesian item response theory (IRT) model [58]. IRT models are commonly used in the social sciences to measure latent characteristics of test takers, such as knowledge, while accounting for variation in test questions.

We analysed accuracy of each identification on a four-point ordinal scale: 0 = incorrect or skipped, 1 = family correct, 2 = genus correct, 3 = species binomen correct. For example, a puff adder (*Bitis arietans*) misidentified as another species in the genus, such as a Gaboon viper (*Bitis gabonica*), would be recorded as a 2, a misidentification as another species in the family Viperidae, such as West African Saw-scaled Viper (*Echis ocellatus*) would be recorded as a 1, and skipping the question or selecting a species outside the family Viperidae would be recorded as a 0.

We used a specific type of IRT model called a graded response model (GRM) that is appropriate for analysing tests scored on an ordinal scale of this type [59]. The GRM models the probability of accurately identifying each photo at the four levels of accuracy described above using a two-parameter logistic distribution. The first parameter sets the centre of the distribution and represents the average accuracy, the second parameter models the spread of the distribution and is known as the 'discrimination' parameter [60]. In the context of the snake identification challenge, the discrimination parameter distinguishes photos or species identified equally well by novices and experts alike—i.e. low discrimination—from photos that are more accurately identified by experts—i.e. high discrimination.

We included group-level terms (aka 'random effects') in the models to account for individual variation in participant knowledge, photo difficulty and species difficulty. Variation in the discrimination parameter was modelled with group-level effects by photo and species.

We fit six candidate models with different combinations of fixed effects (table 1). Three fixed effects were included in all of the models: the region where the photo was taken (which was also shown to the test takers during the survey), the snake family and, for each photo, the number of previous photos a user had seen of the same species—the 'taxa repeat' covariate. The last covariate accounted for the fact that users may have improved their ability to identify species as the test went on. Candidate models included combinations of two additional covariates: photo quality (high or low) and home region (true or false), i.e. whether the photo was from the same region as the user's region. Finally, we also considered models with a home region by photo region interaction term, i.e. allowing the home region effect to vary by global region.

After fitting the six candidate models, we compared models based on approximate leave-one-out (LOO) cross-validation [58,61]. In addition, we evaluated the ability of each model to predict the scores in the held-out data. We summarized prediction accuracy with Bayesian R-squared and mean absolute error. We choose the model with the lowest LOO score for further evaluation.

**Table 1.** Summary of Bayesian IRT model parameters. Each row gives the median, lower and upper 95% credible intervals of the posterior for each fitted parameter in the IRT model. For each parameter, Bulk_ESS and Tail_ESS are effective sample size measures, and Rhat is the potential scale reduction factor on split chains (at convergence, Rhat = 1).

| parameter | estimate | Est.Error | l–95% CI | u-95% CI | Rhat | Bulk_ESS | Tail_ESS |
|---|---|---|---|---|---|---|---|
| group-level (aka 'random') effects: | | | | | | | |
| ~Participant ID (number of levels: 131) | | | | | | | |
| sd(Intercept) | 1.00 | 0.00 | 1.00 | 1.00 | 1.00 | 4000 | 4000 |
| ~Image (aka 'item') (number of levels: 947) | | | | | | | |
| sd(difficulty) | 0.65 | 0.04 | 0.57 | 0.74 | 1.02 | 301 | 750 |
| sd(discrimination) | 0.21 | 0.01 | 0.18 | 0.23 | 1.00 | 2145 | 3278 |
| cor(diff. × disc.) | 0.70 | 0.05 | 0.61 | 0.80 | 1.00 | 1657 | 2342 |
| ~Species (number of levels: 104) | | | | | | | |
| sd(difficulty) | 0.62 | 0.07 | 0.50 | 0.77 | 1.01 | 536 | 1755 |
| sd(discrimination) | 0.31 | 0.03 | 0.27 | 0.36 | 1.00 | 1610 | 2540 |
| cor(diff. × disc.) | 0.34 | 0.14 | 0.04 | 0.58 | 1.01 | 440 | 831 |
| population-level (aka 'fixed') effects: | | | | | | | |
| intercept [1] | −2.36 | 0.49 | −3.36 | −1.43 | 1.01 | 585 | 1485 |
| intercept [2] | −1.17 | 0.48 | −2.11 | −0.24 | 1.00 | 680 | 1639 |
| intercept [3] | −0.53 | 0.47 | −1.47 | 0.38 | 1.00 | 726 | 1701 |
| discrimination | 0.15 | 0.07 | 0.02 | 0.28 | 1.02 | 245 | 701 |
| Colubridae | −1.97 | 0.46 | −2.92 | −1.11 | 1.00 | 811 | 1511 |
| Cylindrophiidae | −2.80 | 0.79 | −4.36 | −1.27 | 1.00 | 1648 | 2545 |
| Elapidae | −1.25 | 0.47 | −2.19 | −0.37 | 1.00 | 968 | 1708 |
| Lamprophiidae | −2.46 | 0.50 | −3.46 | −1.49 | 1.01 | 1121 | 1609 |
| Leptotyphlopidae | −1.58 | 0.77 | −3.14 | −0.09 | 1.01 | 2027 | 2997 |
| Pythonidae | −0.15 | 0.57 | −1.28 | 0.95 | 1.00 | 1259 | 2086 |
| Typhlopidae | −0.81 | 0.77 | −2.32 | 0.72 | 1.00 | 2073 | 2985 |
| Viperidae | −1.09 | 0.46 | −2.02 | −0.24 | 1.00 | 906 | 1627 |
| Asia | −0.09 | 0.19 | −0.44 | 0.28 | 1.00 | 1079 | 1613 |
| Australasia | −0.26 | 0.29 | −0.84 | 0.28 | 1.00 | 1478 | 2448 |
| Europe | −0.04 | 0.22 | −0.45 | 0.40 | 1.00 | 1440 | 2285 |
| NorthAmerica | −0.00 | 0.19 | −0.36 | 0.38 | 1.00 | 1064 | 1921 |
| SouthAmerica | 0.30 | 0.23 | −0.15 | 0.76 | 1.00 | 1165 | 1992 |
| home_region = T | 3.14 | 0.23 | 2.73 | 3.61 | 1.01 | 329 | 1345 |
| taxa_repeat | 0.08 | 0.01 | 0.07 | 0.09 | 1.01 | 410 | 1033 |
| Asia: home_region = T | −2.17 | 0.20 | −2.58 | −1.82 | 1.01 | 475 | 1620 |
| Australasia: home_region = T | −0.51 | 0.54 | −1.47 | 0.68 | 1.00 | 3433 | 3629 |
| Europe: home_region = T | −2.32 | 0.21 | −2.77 | −1.94 | 1.01 | 414 | 1504 |
| NorthAmerica: home_region = T | −1.73 | 0.18 | −2.09 | −1.41 | 1.01 | 569 | 2289 |
| SouthAmerica: home_region = T | −1.63 | 0.29 | −2.21 | −1.06 | 1.00 | 1544 | 2694 |

We fitted the Bayesian IRT models using the 'brms' package in R [58,59,62,63]. A detailed description of model structure, prior specification and R code for replicating the analyses are given in electronic supplementary material, appendix A. Data and relevant code for this research work are stored in GitHub: https://github.com/akleinhesselink/snapp/releases/tag/1.0 and have been archived within the Zenodo repository: http://doi.org/10.5281/zenodo.4288984.

**Table 2.** Accuracy (±95% CI) of participants who identified at least 80 images ($N = 250$). MIVS = medically important venomous snakes.

| | species-level accuracy | genus-level accuracy | family-level accuracy | MIVS or not? |
|---|---|---|---|---|
| 1st quintile (top 20%) | 68 ± 3% | 79 ± 3% | 89 ± 2% | 86 ± 3% |
| 2nd quintile | 45 ± 3% | 62 ± 3% | 82 ± 2% | 68 ± 4% |
| 3rd quintile (middle 20%) | 30 ± 2% | 44 ± 2% | 72 ± 2% | 52 ± 3% |
| 4th quintile | 22 ± 2% | 35 ± 2% | 62 ± 2% | 42 ± 3% |
| 5th quintile (bottom 20%) | 12 ± 2% | 20 ± 3% | 41 ± 4% | 29 ± 4% |
| Whole community | 35 ± 3% | 48 ± 3% | 69 ± 3% | 55 ± 3% |

# 3. Results

## 3.1. Challenge summary

A total of 1027 participants submitted 117 897 unique IDs over a period of 6 days (1–1088 IDs/participant, mean ± s.e.m. = 114 ± 7 IDs/participant, median = 13 IDs/participant). Participants came from 49 countries primarily in North America (50%) and Europe (22%), but also Africa (10%), Asia (including Middle East; 10%), South America (3.5%) and Australia/Oceania (3.5%). Just over 53% of first-time visits were from Facebook (37%) and other social media platforms (9%) or from direct entry (30%; i.e. no external referrer). Search engine traffic (4%) and website referral (2%) were other, less important channels for recruiting participants. Two hundred and fifty participants (24% of all participants) identified more than 80 photos each (103 160 IDs, 88% of all IDs) and this subset was used in statistical analyses because this threshold provided a reasonable balance between the number of participants and the number of photos per participant.

In addition to the four-point scale used for analysis, we also scored whether identifications were correct or incorrect with respect to medical importance (MIVS or not; table 2), using the 2008 World Health Organization Category 1 and Category 2 MIVS species [64], updated to reflect new species descriptions and taxonomic changes over the past 12 years [15]. Overall, 55 ± 3% of identifications were correct in terms of the MIVS status of the species (top quintile: 86 ± 3%, bottom quintile: 29 ± 4%; table 2).

At least 72 IDs were submitted during every hour of every day (mean = 814 ± 86 [95% CI], max = 2794 IDs per hour) throughout the 6-day challenge. The median time taken by participants to submit their ID was 17 s, and 95% of participants submitted their ID within 125 s. There was not significant variation in speed among snake taxonomic groups, global regions, photo quality or whether answers were correct or incorrect at any taxonomic level (electronic supplementary material, table S3).

## 3.2. Identification

In total, the 250 participants who identified at least 80 photos submitted 44 696 IDs that were correct at the species level, 12 471 at the genus (but not species) level, 19 524 that were correct at the family (but not genus) level, 13 261 that were incorrect at the family level and 13 208 that were skipped. We collected 68–157 IDs per photo, of which 4–94% were correct at species level (10–99% correct at genus level). There were between 5 and 47 unique responses per photo, and the number of unique responses per photo was negatively correlated with the percentage of those responses that were correct (mean ± s.d. = 10 ± 3 when greater than 75% were correct, 25 ± 8 when less than 25% were correct; electronic supplementary material, figure S1); however, some photos with few unique responses nevertheless were mostly identified incorrectly. The most extreme case was an photo of *Crotalus scutulatus* that was misidentified as *Crotalus oreganus* 72% of the time (57 of 79 IDs) and correctly identified 11% of the time (9 of 79 IDs) despite having only eight unique responses.

Out of 963 photos analysed, 336 (35%) were correctly identified at the species level by the majority of participants. By contrast, for only 16% of the photos was the most common answer an incorrect identification. Only six photos (less than 1%) were correctly identified at the species level by more than 90% of participants, and 25 photos (2.6%) were correctly identified at the species level by less than 10% of participants (mostly by participants in the top quintile; 70% versus 51% for other photos; $X^2 = 13.53$, $p = 0.009$).

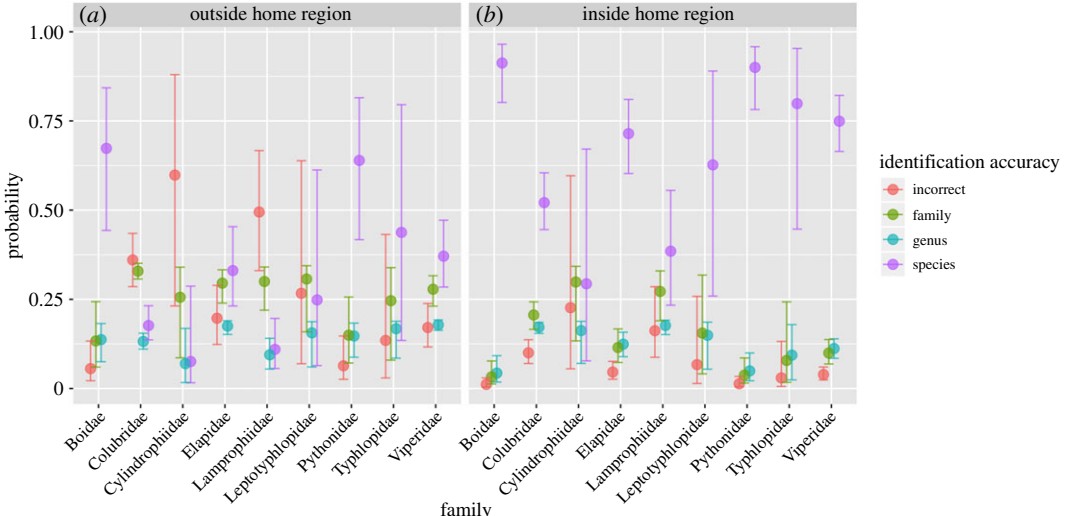

**Figure 2.** Estimated probabilities of correctly identifying snakes in nine families at the family, genus and species level. (*a*) Gives the probabilities of identifying a snake from outside of the participant's home region, and (*b*) gives the probability for snakes from within the participant's home region. For all families, the reference region is North America and the estimates refer to the probability of correctly identifying the first image of a particular species shown to a participant. Points and error bars show the back-transformed median and 95% Bayesian credible intervals, respectively, derived from the Bayesian IRT model.

A small percentage of the photos that we used were misidentified at the source (1.5% of iNaturalist photos and 1.3% of photos from Facebook snake groups) and were not corrected before being included in the challenge, although many participants managed to correctly identify these photos anyway and reported the mistakes using a feedback form built into the interface. These misidentifications have been fixed throughout the dataset except in the very few cases where no definitive ID could be reached; such photos were dropped from analyses.

## 3.3. IRT model selection and summary of top model

After fitting the IRT models, we checked convergence of the sampling algorithms by visual inspection of the sampling chains and with the split-chain Rhat statistic. We also tested whether model predictions were appropriate with graphical posterior predictive checks (see electronic supplementary material, appendix A for more details). The top IRT model had a ΔLOOIC score of 36 and contained fixed effects terms for family, photo region, home region, the interaction between photo region, home region and taxa repeat (see electronic supplementary material, table S4 for full model comparison). The out-of-sample R-squared for the model was 0.47 or 0.36 depending on whether group-level effects were used (electronic supplementary material, table S4). Mean absolute error for out-of-sample predictions was 0.83 or 1.17 score units depending on whether group-level effects were used.

Of the fixed effects, variation in identification accuracy among families exceeded that among global regions by nearly sixfold (table 1). Participants were much more able to correctly identify photos from their home region than those outside of their home region (table 1). There was a small but significant positive effect of the number of previous photos a user had seen of the same species (table 1). Additional plots of fixed effects are available at https://beautiful.shinyapps.io/snake_id_by_region/.

## 3.4. Effect of taxonomic family

Photos of pythons (58%) and boas (58%) were the most likely to be correctly identified to species, whereas photos of snakes from more obscure families (e.g. lamprophiids) were the least likely to be correctly identified to species (27%; figure 2). This is despite the fact that some lamprophiids are MIVS (i.e. the genus *Atractaspis*, which has historically had perhaps the most unstable family-level placement of any snake genus [65], having been a member of Viperidae, Elapidae, Colubridae, Atractaspididae and Lamprophiidae). Vipers (Viperidae) were the most likely to be recognized at the family level (figure 2)—84% of identifications were correct at the family level, compared with 69% at the genus level and 52% at the species level. Across taxa, correct identifications were much more

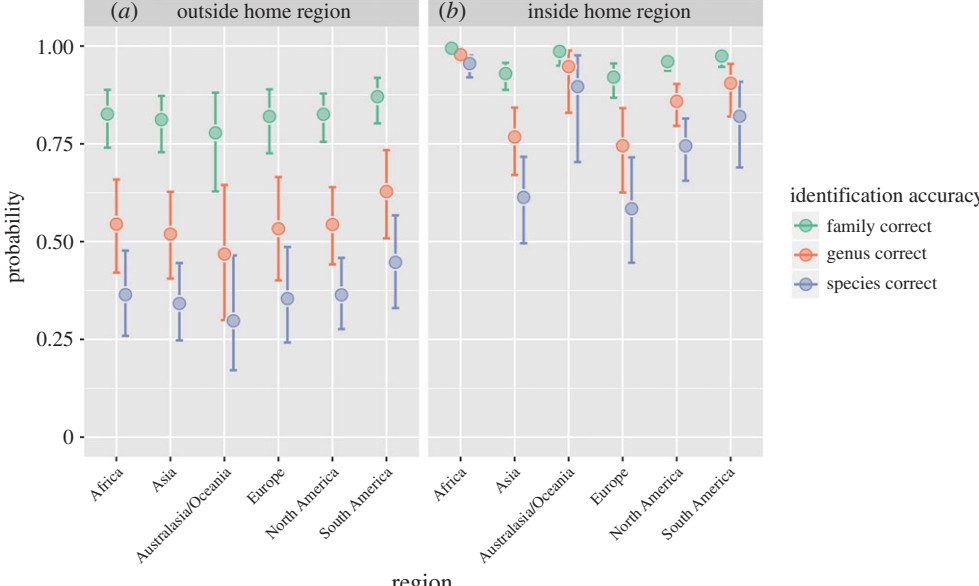

**Figure 3.** Estimated probabilities of correctly identifying snakes in six global regions at the family, genus and species level. (*a*) Gives the probabilities of identifying a snake from outside of the participant's home region, and (*b*) gives the probability for snakes from within the participant's home region. For all regions, the reference family is Colubridae and the estimates refer to the probability of correctly identifying the first image of a particular species shown to a participant. Points and error bars show the back-transformed median and 95% Bayesian credible intervals respectively derived from the Bayesian IRT model. This plot can be viewed using different reference families at https://beautiful.shinyapps.io/snake_id_by_region/.

frequent (94% at the family level) when only the top 20% of participants were considered. There were large uncertainty estimates associated with families represented by only a few species in the challenge (Cylindrophiidae, Typhlopidae and Leptotyphlopidae).

## 3.5. Effect of global region and participant home region

When participants were asked to identify snakes outside of their home region, identification rates across global regions were more or less similar (figures 2 and 3). There was a strong positive effect of home region, i.e. users were more able to correctly identify snakes from their own region (table 1; figures 2 and 3). However, there was a significant home region by global region effect such that the home region effect was strongest for Africa and Australia/Oceania (figure 3) and weakest for Europe.

## 3.6. Effect of medical importance

Because there are very few families with both MIVS and non-MIVS, the effect of MIVS was strongly confounded with the effect of family, which limited our ability to test them both simultaneously in the IRT model. Comparing the raw data instead, we found that medically important genera were mistakenly identified as harmless genera (false-negative) between 1% and 20% of the time (table 3), depending on the genus. Harmless genera were mistakenly identified as medically important genera (false positives) between 0% (*Charina*, *Rena*) and 42% (*Pliocercus*) of the time, depending on the genus (mean = 6%).

## 3.7. Effect of species diversity

We investigated whether the difficulty of identifying snakes increased with the number of species in each family and genus. Using a simple linear regression, we found no support for a relationship between diversity within a genus or a family and identification accuracy ($R^2 = 0.03$ for genus, $R^2 = -0.04$ for family). Several very small genera were frequently misidentified at the species level (e.g. greater than 85% of the time for *Duberria*, *Grayia*, *Hapsidophrys*, *Pliocercus*, *Scaphiodontophis* and *Sinonatrix*), and smaller families were among both the most (e.g. Cylindrophiidae) and least (e.g. Boidae, Pythonidae) frequently misidentified.

**Table 3.** Medically important venomous snake (MIVS) genera and the genera with which they were most commonly confused.

| family | genus | no. species | number of responses | per cent correct (genus/species level) | per cent identified as non-MIVS | most frequently chosen incorrect genus (%) | |
|---|---|---|---|---|---|---|---|
| | | | | | | MIVS | non-MIVS |
| Colubridae | Thelotornis | 4 | 882 | 53.9%/49.4% | 13.6% | Dispholidus (2.2%) | Oxybelis (4.6%) |
| | Dispholidus | 1 | 854 | 41.7%/41.2% | 15.3% | Dendroaspis (8.7%) | Philothamnus (2.6%) |
| Elapidae | Micrurus | 80 | 2081 | 70.4%/41.8% | 12.9% | Micruroides (0.8%) | Lampropeltis (2.5%) |
| | Naja | 28 | 3020 | 70.3%/41.5% | 4.0% | Ophiophagus (1.5%) | Ptyas (1.4%) |
| | Dendroaspis | 4 | 1430 | 70.0%/53.9% | 6.6% | Dispholidus (1.3%) | Philothamnus (2.3%) |
| | Laticauda | 8 | 1027 | 68.9%/50.7% | 2.6% | Bungarus (7.0%) | Emydocephalus (0.5%)[a] |
| | Acanthophis | 8 | 960 | 60.5%/53.8% | 2.3% | Notechis (7.1%) | Telescopus (0.4%) |
| | Ophiophagus | 1 | 942 | 56.0%/55.7% | 5.4% | Naja (11.0%) | Boiga (1.3%)[b] |
| | Bungarus | 15 | 653 | 41.6%/21.2% | 17.9% | Ophiophagus (1.2%) | Lycodon (4.1%) |
| | Notechis | 1 | 939 | 40.6%/39.9% | 5.9% | Pseudechis (8.1%) | Telescopus (1.6%) |
| | Pseudonaja | 9 | 918 | 32.1%/23.3% | 9.4% | Pseudechis (11.2%) | Furina (0.9%)[c] |
| | Pseudechis | 9 | 979 | 31.6%/29.3% | 5.3% | Pseudonaja (16.4%) | Boiga (1.2%)[b] |
| Lamprophiidae | Atractaspis | 21 | 882 | 47.1%/38.3% | 20.6% | Naja (1.8%) | Pseudaspis (2.0%) |

(*Continued.*)

**Table 3.** (Continued.)

| family | genus | no. species | number of responses | per cent correct (genus/species level) | per cent identified as non-MIVS | most frequently chosen incorrect genus (%) | |
|---|---|---|---|---|---|---|---|
| | | | | | | MIVS | non-MIVS |
| Viperidae | Crotalus | 47 | 4135 | 87.3%/61.9% | 2.1% | Sistrurus (1.7%) | Nerodia (0.5%) |
| | Agkistrodon | 6 | 2035 | 85.3%/75.5% | 3.0% | Crotalus (1.6%) | Nerodia (1.6%) |
| | Bothriechis | 11 | 986 | 75.7%/71.7% | 1.3% | Bothrops (1.2%) | Corallus (0.2%) |
| | Lachesis | 4 | 996 | 74.3%/71.9% | 0.8% | Bothrops (6.3%) | Xenodon (1.0%) |
| | Bitis | 18 | 865 | 74.1%/64.0% | 1.2% | Echis (1.8%) | Causus (0.6%)[d] |
| | Trimeresurus | 46 | 2094 | 72.4%/37.2% | 2.8% | Protobothrops (1.8%) | Ahaetulla (0.4%) |
| | Bothrops | 45 | 1933 | 67.7%/42.0% | 2.8% | Lachesis (1.9%) | Boa (0.2%) |
| | Vipera | 24 | 1252 | 66.5%/50.5% | 12.9% | Bitis (0.5%) | Natrix (7.3%) |
| | Sistrurus | 3 | 1007 | 61.6%/41.5% | 7.8% | Crotalus (13.5%) | Nerodia (2.1%) |
| | Daboia | 4 | 918 | 60.7%/59.9% | 9.5% | Naja (1.9%) | Python (4.2%) |
| | Echis | 10 | 1163 | 56.2%/34.3% | 4.3% | Bitis (4.5%) | Causus (1.6%)[d] |
| | Gloydius | 16 | 1104 | 44.2%/28.0% | 4.4% | Daboia (3.2%) | Python (0.4%) |

[a]Emydocephalus are elapid sea snakes with essentially non-toxic venom and vestigial, non-functional venom glands [66,67].

[b]Boiga are large, rear-fanged colubrids that are capable of delivering pediatric bites for which medical treatment is required, but are generally not considered MIVS [68–70].

[c]Furina are terrestrial Australian elapid snakes that possess functional fangs and venom glands, but are not regarded as dangerous to humans, although their bites are said to be very painful and some species can produce serious sequelae [71–73].

[d]Causus are small viperids with functional fangs and venom glands, but are not considered MIVS because bites, though painful, are not life-threatening and are unlikely to cause serious ill effects [74,75]. Other rear-fanged non-MIVS genera listed here include Oxybelis, Telescopus, Xenodon and Ahaetulla; bites from these snakes can cause mild to moderate transient, local effects and occur almost exclusively in captivity [39].

(a)    *Boa constrictor* (top 20%)    (b)    *Boa constrictor* (bottom 20%)

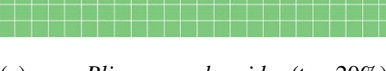
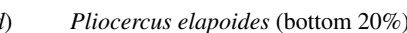

(c)    *Pliocercus elapoides* (top 20%)    (d)    *Pliocercus elapoides* (bottom 20%)

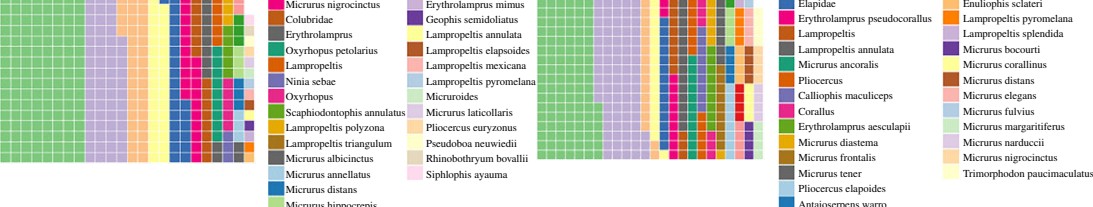

**Figure 4.** Waffle plots of correct and incorrect identifications (derived from raw data) of the top 20% and bottom 20% of users who identified at least 80 images for the most frequently correctly identified species (*Boa constrictor*) and the least frequently correctly identified species (*Pliocercus elapoides*) (see the electronic supplementary material, table S2 for exact percentages).

## 3.8. Variation among species

Across all participants and species, *Boa constrictor* was the most accurately identified (80% correct at the species level; figure 4). Among MIVS, New World pit vipers in the genera *Bothriechis*, *Crotalus* and *Lachesis* were the most accurately identified, whereas kraits (*Bungarus*), stiletto snakes (*Atractaspis*) and night adders (*Causus*) were the least likely to be accurately identified, often being confused with non-venomous snakes (maximum 19% of the time in the case of *Bungarus* versus minimum 1.3% of the time for *Bothriechis*; table 3). The most frequently misidentified species was *Pliocercus elapoides*, a coralsnake mimic [76], which was misidentified as a venomous coralsnake 44% of the time and was correctly identified at the species level only 12.8% of the time (figure 4). Conversely, the species of coralsnake it most closely resembles, *Micrurus diastema*, was misidentified as harmless 29% of the time (usually as similarly patterned species in the genera *Pliocercus*, *Erythrolamprus*, *Lampropeltis* or *Sonora*).

## 3.9. Effect of photo quality

The effect of photo quality (i.e. focus, lighting, angle, proportion of the frame occupied by the snake) was not included in our top model (electronic supplementary material, table S4). However, examination of the random intercepts fitted in the IRT model showed that photos we rated 'low-quality' were generally more difficult to identify than photos we rated as 'high-quality' (electronic supplementary material, figure S2). We suspect that the taxa repeat effect (the number of previous photos of the same species the participant had seen) may have masked the photo-quality effect because by default low-quality photos were shown after high-quality photos.

## 3.10. Participant learning over time

Even though our challenge was of short duration, many participants expressed the view (via the forum) that it helped them gain valuable experience identifying snakes from taxonomic groups or regions of the world where they were weak. Across all participants who identified 10 photos of at least one species, the

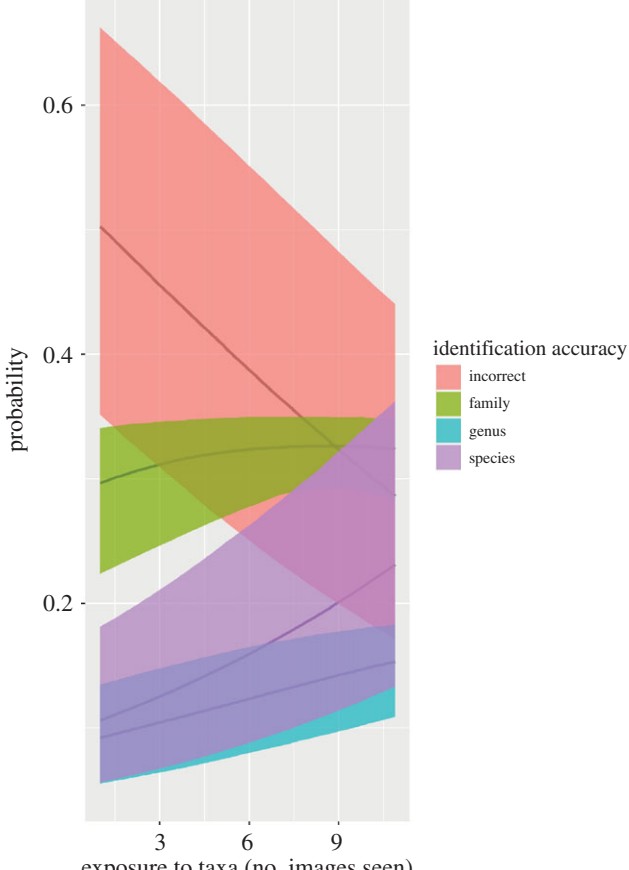

**Figure 5.** Increases in rate of correct identifications at the family, genus and species level as participants saw more images of the same species. The reference region is North America, and the reference family is Colubridae. Lines and shaded regions show the back-transformed median and 95% Bayesian credible intervals, respectively, derived from the Bayesian IRT model.

raw percentage of photos correctly identified at the species level improved from 51% when a species was first encountered to 59% the 10th time it was encountered. The improvement in accuracy over time was quantified in the IRT model by the taxa repeat effect (figure 5). As an example, the average participant outside of North America would have a $17 \pm 2\%$ probability of correctly identifying the first photo they saw of a North American colubrid but a $38 \pm 4\%$ probability of identifying the 10th photo of that same species (Figure 5). Gains were more modest for family region pairs with higher or lower baseline accuracy (e.g. from $74 \pm 10\%$ to $88 \pm 5\%$ for South American boids, from $11 \pm 3\%$ to $23 \pm 6\%$ for African lamprophiids, among users from outside those regions).

## 3.11. Species and photo difficulty

The hierarchical intercepts for species and photo fitted by the IRT model provide estimates for the difficulty of each species and photo in the test. Based on these, taking all other covariates into account, from most difficult to easiest, the five most difficult species for users to identify were *Psammodynastes pulverulentus*, *Malpolon monspessulanus*, *Pliocercus elapoides*, *Cylindrophis ruffus* and *Chrysopelea ornata* (electronic supplementary material, figure S2). The five least difficult species were *Python sebae*, *Bitis arietans*, *Naja melanoleuca*, *Dendroaspis angusticeps* and *Boa constrictor* (electronic supplementary material, figure S2).

The hierarchical nature of our model allowed us to estimate the standard deviation in user knowledge or ability on the same scale as photos and species difficulty. Comparing these estimates, we found that the standard deviation associated with participant ability was 1.6 times greater than the standard deviation associated with species difficulty and 1.5 times greater than the standard deviation of photo difficulty within species (table 1).

## 3.12. Discrimination by species and photo

The discrimination parameter in our IRT model varied widely across species and more modestly across photos within species (electronic supplementary material, figure S3). Species and photos with high discrimination ability are useful in telling apart users that are very skilled from those who are less skilled. There was a moderate positive correlation between discrimination and difficulty of individual photos ($r = 0.70$, 95% CI = 0.61–0.80), and a weak correlation between discrimination and difficulty of species ($r = 0.34$, 95% CI = 0.04–0.58). This was reflective of the fact that species with low discrimination estimates included both frequently misidentified (e.g. *Psammodynastes pulverulentus*) and some frequently correctly identified species (e.g. *Boa constrictor*).

## 4. Discussion

We used an online open identification challenge to assess the capacity of online communities to correctly identify snakes from photographs. We found that accuracy varied significantly among snake families, with taxa from less diverse families such as Boidae and Pythonidae being the most readily identified and those from more diverse families, such as Colubridae, and more obscure families, such as Cylindrophiidae, being most difficult to identify. Participants were also more able to identify snakes found within their own home region rather than snakes from other global regions. This last effect was strongest for snakes from Africa and Australia/Oceania. However, even after controlling for the above effects, identification accuracy still varied considerably across individual participants, species and photographs. Variation in participant knowledge or ability was the largest source of variation among these. The most skilled participants (top 10 of 250 analysed) were able to identify 81% of photos correctly at the species level, 89% at the genus level and 95% at the family level, whereas the least skilled participants (bottom 10 of 250 analysed) only identified 6% of photos correctly at the species level, 12% at the genus level and 23% at the family level. Average accuracy also declines quickly as more participants are added to the top tier: even the top 20% of participants analysed ($N = 50$) were only able to correctly identify snakes at the species level 69% of the time (table 2).

The speed with which the participants submitted identifications conforms to the hypothesis that experts can make visual identifications of diverse categories of objects by gestalt, rather than using taxonomic keys or other more quantitative features [77]. Many people with expertise in snake identification express that they cannot easily describe how they know the identity of a certain species, particularly in photos of poor quality, but that they simply 'just know'. Narrowing the number of potential species down based on geographical location is a critical part of snake identification [78], which we did not explicitly test here (participants always had the relevant geographical location available, although higher familiarity with the fine-scale distribution of a species probably plays into the strong home region effect). Although body size, body shape, habitat, behaviour, colour pattern and unique features such as rattles [79], hoods [80] or thermal pits [81] all play a role, further research is needed to determine the relative importance of these and other characters (especially colour and pattern) in snake identification, particularly since many taxonomically relevant features are not easily photographed in the field [6]. For instance, species descriptions frequently rely on counts of the number of scales on specific regions of a snake's body, a feature that is rarely visible even in high-quality photographs and often requires an expert to handle the snake (e.g. [82]). Finally, the pace of our short-duration challenge with thousands of photos surely differs from more realistic scenarios in which an expert is sent one or a few photos of a specimen to identify at a time. In contrast to our competitive scenario, where participants earned points for identifying more photos, experts who are aware that their identification is scientifically or medically important are likely to want to be more certain, taking the time to consult resources if the identification is not straightforward.

The probability of getting a correct identification did not vary across regions when the participant was not from that region (figure 3). Baseline accuracy for participants varied among snake family, with the general pattern that large constrictors (boas and pythons) were the easiest, followed by vipers, elapids, colubrids and lamprophiids. There was not enough information to say whether other families were harder or easier because there were fewer photos and species. The core 100 species were selected for balance among global families and regions in order to reflect realistic scenarios, but more targeted experiments could elucidate their relative difficulty. We also suggest more explicitly testing the effects of the diversity of species in different families, genera or global regions using photo datasets fine-tuned to answer these questions.

We found that participants had a substantial advantage when identifying snakes from their home region over participants that came from other regions (figures 2 and 3). The home region advantage was the strongest in Africa and Australia, suggesting that participants outside of these regions are especially unfamiliar with the snakes that occur there compared with participants who live there, and/or that the participants we managed to recruit from these regions were particularly skilled. Relatively few experts are familiar with all snakes worldwide, and most snake species are endemic to a single global region. Intuitively, people interested in snakes are more likely to be familiar with taxa that they routinely encounter in the field and read about in local or regional field guides and journals. This emphasizes both the difficulty of the task of snake identification for international healthcare workers (who may lack both regional experience and motivation to learn about snakes) and the need to involve community members with local and regional expertise in healthcare systems in all parts of the world, both online and off. In addition, consensus that weighs the empirically assessed (not self-assessed) level of expertise of each contributor may be advantageous when attempting to crowdsource an identification. Although we did not have the granularity to look at introduced species, the growing number of introduced populations of snakes outside their native range suggests that out-of-range identifications will become increasingly relevant [83,84].

We now have a sufficiently sophisticated understanding of snake evolution to be confident that not all venomous snakes are close relatives [85], and the potential medical importance of some obscure lineages historically thought to be harmless is becoming better understood [39]. Thus, even higher-level taxonomic identifications at the genus or family level may have utility to healthcare providers in making decisions about treating snakebite. Although species-level identification accuracy was generally low, the top 20% of participants correctly identified $80 \pm 3\%$ of photos at the genus level and $90 \pm 2\%$ at the family level (table 2). Genus-level identification is always sufficient for deciding whether a snake is an MIVS or not, and is usually sufficient for antivenom choice (although there are notable exceptions; e.g. [86,87]). Confusion of similar species frequently occurs within a family (table 3) and many of the most difficult species in our dataset come from families that are not widespread (e.g. Cylindrophiidae), mimic species in other families (e.g. *Pliocercus*), or have proven difficult to classify at the family level (e.g. *Psammodynastes* [88]). Furthermore, we recommend that when an identification is given at the genus or family level based on past phylogenetic understanding but no longer reflective of current taxonomy (e.g. calling *Atractaspis* a viperid or lumping *Mixcoatlus* with *Porthidium* or *Ophryacus*), a more extensive set of data such as ours should be used in combination with machine learning [89,90] to flag taxa that are easily mistaken or commonly misidentified.

The low number of photos that were misidentified at the source (1.3–1.5% by the crowdsourcing processes active on iNaturalist and in Facebook groups) suggest that crowds working together can achieve higher accuracy than individual crowd members working alone [91], although social influence may undermine the wisdom of the crowd in some circumstances [92,93]. At least one participant correctly identified every photo to the species level, and 336 photos (35%) were correctly identified at the species level by the majority of participants. In more realistic scenarios, participants could have communicated with one another and eventually agreed upon an identification, similar to the open identification system of iNaturalist.org and the semi-open ones of eBird (ebird.org/) and HerpMapper.org (which use regional data experts to validate or suggest corrections to observations [51,94]). However, some photos (16% where the most common answer is incorrect) would require more advanced aggregation techniques such as integrating participant expertise or reputation [95] to avoid accepting an incorrect consensus.

Combining crowdsourcing with artificial intelligence into human–machine systems for snake recognition [89] has the potential to produce results superior to those of either one alone through smart task allocation (e.g. humans could pick from a list of species deemed likely by the AI, or suggest a species not on the list). Large labelled datasets are essential for training neural networks, and these can be produced more efficiently by crowds than by experts. However, healthcare providers and patients are likely to prefer a 'human in the loop' approach [96] to snake identification that combines crowdsourcing and vetting by experts.

Borrowing from the testing and social science literature, we used an IRT model to estimate how well particular snake taxa and particular photographs were able to discriminate between participants with high and low identification accuracy (electronic supplementary material, figure S3). Questions with a high discrimination ability are favoured in the design of standardized tests because they increase the overall reliability of the test scores [60]. In the context of online snake identification, species with a high discrimination ability should be included in tests designed to reliably vet participants' abilities to accurately identify snakes from photos online. As our results show that variation in participant

knowledge is significant (table 1), such a vetting process would be critical for building the trust of medical professionals in any crowdsourced online snake identification platform.

Many of the most medically important venomous snakes are part of complex groups of often very similar species (e.g. *Bungarus*, *Echis*, Asiatic *Naja*). The systematics of many such complexes, and the definition and identification of the species therein, are the subject of considerable debate [37]. Although we attempted to provide all reasonable taxonomic names, recent taxonomic changes were responsible for a further 1.3% of 'incorrect' identifications, which we painstakingly corrected after data collection using a combination of the R package taxize [97], literature searches and expert consultation. This was sufficiently labour-intensive that it would not be realistic at large scales. The problem of taxonomic instability can have significant public health implications [98,99] and effort should be taken during taxonomic revisions of MIVS to maximize stability [100–103], particularly because antivenom databases do not currently cross-list well with older taxonomic names. In particular, the splitting up of widespread species based on inappropriate sampling, especially of contact zones or methodology, e.g. 'isolation by sampling gap' and coalescent approaches, should be avoided [16]. The use of vague or out-of-date names is widespread; in a review of 150 publications published since 1978 where 8885 snakes in snakebite cases were identified [18], a substantial number of individuals were reported using only the common name (21%), had been moved to a different genus (1%), had experienced other taxonomic changes (e.g. splits or lumps; 2.7%) or contained minor misspellings (less than 1%). Similarly, Wüster & McCarthy [37] found that only 21% and 46% of toxicological and clinical publications provided sufficient information to confidently identify the species of Asiatic *Naja* or *Echis carinatus* complex, respectively, involved (both groups have major differences in venom composition, symptomology and antivenom efficacy among species [87,104]).

The rate at which new names are adopted by professional and avocational snake enthusiast communities probably differ, but have not been measured. A frequently updated global database of snake range maps (presently in development; [15,105]) combined with an algorithm for narrowing down choices based on the geographical origin of an identification request would help guide both crowd members and clinicians to the correct name (a good example of this is the filters used to guide users to which bird [50] or plant [106] species are expected at a particular date and location). We suggest that more regular interactions among herpetologists, citizen science identifiers and clinicians have the potential to decrease these errors and improve articulation of the correct name with a particular case by decreasing ignorance of or disregard for the latest widely accepted taxonomic changes [37].

Although at first glance participants in our challenge appear to have improved their accuracy over time (figure 5), we suggest that this phenomenon could be partly explained by the limited number of core species used in the challenge (100) and the fact that participants were given immediate feedback on whether an identification was correct and what the correct identification was. Even though the order of presentation was haphazard, some participants used the forum to admit that they had caught on to the pattern and suggested adding more 'distractor species' in future iterations. In more realistic scenarios, this phenomenon could lead to misidentification of rare species as more common species.

Within any given species of snake, we found that 'low-quality' photos were typically more difficult to identify than 'high-quality' photos; however, the inclusion of photo quality was not supported in our top model. This may reflect features of the online challenge design: first, participants were able to select whether they wanted to identify high- or low-quality photos and participants who elected to identify low-quality photos may have been above a certain skill threshold and/or moved on to low-quality photos after they had identified all high-quality photos. Indeed, the number of high- and low-quality photos identified by participants in the top quintile was nearly equal, whereas the numbers of low-quality photos identified by participants in the other quintiles were much lower (between 25% and 75% as many) than the numbers of high-quality photos, and participants who identified more than 500 photos provided data on a much more balanced sample of photo difficulties (mean ± s.d. = 55 ± 14% high-quality photos [$N = 70$] versus 82 ± 32% for participants who identified between 100 and 500 photos [$N = 153$]). Second, identification ability increased as subjects saw more photos, but if a participant left all default settings unchanged, low-quality photos were only shown after the high-quality photos were shown. Thus, the positive effect of repeat exposure to the same taxa throughout the test may have masked some of the difference between high- and low-quality photos. Finally, it was challenging for us to identify suitably low-quality photos of which we could be certain of the identification. Thus, harder photos (e.g. those that can only be reliably identified to genus or family) surely exist, but they were not included in this study.

Snake identification remains challenging due to the high diversity of species and the high degree of morphological similarity among certain species. Although many professional herpetological organizations exist, which count among their membership numerous experts in snake identification, few healthcare professionals would be likely to join. Several human–snake conflict and snakebite-specific NGOs (e.g. Asclepius Snakebite Foundation (www.snakebitefoundation.org), Save the Snakes (savethesnakes.org), Advocates for Snake Preservation (www.snakes.ngo), Indian Snakebite Initiative (snakebiteinitiative.in)) as well as accredited providers of snake awareness, snakebite first aid and venomous snake handling training (e.g. African Reptiles and Venom (africanreptiles-venom.co.za), African Snakebite Institute (www.africansnakebiteinstitute.com)), might be more useful to the average healthcare professional. However, we suspect that healthcare professionals would probably experience the most favourable cost : benefit ratio in terms of building their snake identification network and capacity by simply joining snake identification communities on iNaturalist, HerpMapper and social media platforms that they are likely to already be using, such as Twitter or Facebook (electronic supplementary material, table S1) [107,108]. Major strengths of these communities include the fact that community members can watch or participate at their discretion, newcomers and novices are frequent and welcomed, the identification process is transparent, discussion of particular identifications is encouraged, and a permanent record of identifications allows for older problems in snake identification to be located and revisited. Community group administrators also curate open collections of reference materials that are available to all group members as public files, containing links to frequently asked questions, diagrams showing salient morphological features, range maps and other such resources. These communities play an important role in increasing the capacity of amateurs and novices worldwide through repeated exposure to identified photos of relevant snake species. Combined with next-generation field guides [109], automated photo recognition [110–112] and vetting by trained experts, crowdsourcing has the potential to augment herpetology (including documentation of biodiversity and identification of rare and threatened species) and its application to a major global health problem, snakebite envenoming, by providing fast, accurate and verifiable identification of snakes from photos.

# 5. Conclusion

Taken together, our results demonstrate the potential of a large and active community of professionals and skilled amateurs to quickly and accurately identify snakes from photos, with only a minor to moderate influence of the geographical origin of the photo or its quality. Photo quality had a relatively small effect on identification accuracy (electronic supplementary material, table S4), suggesting that even relatively poor photos taken with low-resolution cell phone cameras or in the rushed circumstances of a snakebite may be identifiable, at least to skilled experts. Participation in citizen science initiatives, such as HerpMapper and iNaturalist, can act as training programmes, like our snake identification platform, and could be used broadly to build identification capacity. In particular, social media-based 'communities of practice' support dispersed communities and provide forums for learning that are integrated into channels that many people are already using [113]. Participation in online snake-focused communities, mentorship from local herpetologists and skilled avocational snake enthusiasts, and immersive laboratory or field experiences relevant to snake identification are likely to be at least as useful in training healthcare providers and other health professionals to identify snakes as isolated characteristic-based training exercises. Identification by an expert familiar with the local snake fauna will always be the gold standard, and precedent from the field of medical entomology [114] could help establish a role for professional medical herpetologists. Until such time, however, we suggest that collaboration between herpetology and public health, epidemiology and clinical practice is essential, and parties interested in snakebite ought to involve members of the active online amateur and expert snake enthusiast communities more formally and more often, in order to share knowledge, build capacity, and improve snakebite treatment outcomes, in support of the ambitious goals set in 2019 to reduce SBE deaths and disability by half by 2030 [19].

Ethics. We provide a letter from the Swiss Human Research Act (HRA) and Cantonal Ethics Committee Geneva (CCER) confirming that this study was outside the scope of human subjects institutional review board. We also confirm that the research objectives were outlined on the challenge webpage and that Terms of Use as well as a Privacy Policy are linked at the bottom of all pages of snakes.citizenscience.ch. We initially reported that some images came from Twitter, but in fact this was not the case for this study. All metadata have been checked for anonymity and supplementary material is now available on GitHub (data remain available on figshare but we now opt to use GitHub for all data and code).

Data accessibility. Code required to replicate the analyses is available at https://github.com/akleinhesselink/snapp/releases/tag/1.0 and has been archived within the Zenodo repository: https://doi.org/10.5281/zenodo.4288984.

Authors' contributions. A.M.D. designed the experiment, assembled the image set, verified all identifications, analysed much of the data and wrote the paper; A.R.K. designed and implemented the IRT model, created the GitHub repository where data and code are stored and wrote appendix A; A.M.D., I.B. and R.R.d.C. developed the idea, discussed experimental design and constructed the broader context; M.R.M., J.L.F.-M., and F.G. contributed expertise in citizen science; F.G.-J. managed the community; M.T. created and managed the platform front-end; C.G. created and managed the platform back-end; C.E.S. and W.W. provided images, helped β-test and provided herpetological expertise; all authors contributed to writing the manuscript.

Competing interests. We declare we have no competing interests.

Funding. This research was supported by a grant (QS04-20) from the Fondation privée des Hôpitaux Universitaires de Genève. R.R.d.C. was supported by Fondation Louis-Jeantet, by A. Flahault and F. Chappuis at the Institute of Global Health at the Department of Community Health and Medicine of the University of Geneva and the Division of Tropical and Humanitarian Medicine at the University Hospitals of Geneva. M.R.M., J.L.F.-M., F.G.-J., C.G., M.T. and F.G. was partially funded by the European Union's Horizon 2020 research and innovation programme under grant agreement no. 872944.

Acknowledgements. We thank all the participants in our snake identification challenge, especially those who took the time to report errors. P. Uetz provided support for snake taxonomy through the Reptile Database [15]. D. Becker, C. Smith and M. Pingleton provided support for and access to HerpMapper data, including common names of snakes in English. S. Durham and E. Orel provided support with experimental design, data analysis and multinomial regression. C. Montalcini supported data collection and organization. J. Childers, S. Durso, P. Freed, D. Morris, M. O'Shea, A. Schmitz, M. Thomas and J. Vanek β-tested and provided detailed, constructive feedback on the challenge, and M. O'Shea allowed us to give his book as a prize. The administrators and moderators of several Facebook groups, especially 'Snake Identification' and 'Wild Snakes: Education and Discussion' (W. Brekhus, C. Collen, J. Condrey, L. Connelly, W. Dillon, J. Foster, B. Gigi, E. Hobbs, K. Kolodziej, K. Mayle, D. McGuire, S. Miller, R. Morgan, R. Pals, A. Rooks, B. Rulon-Miller, J. Schauer, F. Theart, M. Van Valen, Z. Van Vuuren), helped β-test the challenge, provided detailed, constructive feedback in real time and helped promote the challenge on social media. X. Glaudas, R. Gray, T. Huang, Y. Kalki, J. Louies, K. Mebert, D. Pandey, S. Ruane and W. Wüster donated photos and helped A.M.D. verify identifications. H. Heinz provided useful comments on earlier drafts of the manuscript.

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
