## [Reviewer comments · Royal Society Open Science]

Review History

RSOS-191835.R0 (Original submission)

Review form: Reviewer 1

Is the manuscript scientifically sound in its present form?

Yes

Are the interpretations and conclusions justified by the results?

Yes

Is the language acceptable?

Yes

Do you have any ethical concerns with this paper?

Yes

Have you any concerns about statistical analyses in this paper?

No

Recommendation?

Major revision is needed (please make suggestions in comments)

Comments to the Author(s)

This study used a citizen science web site project with photographs of snakes to assess the ability, availability, speed, and accuracy of participants from around the world to identify snakes, particularly medically important venomous snakes; it also assessed the effects of photograph quality. The authors explain that healthcare practitioners treating snakebite patients are often presented with photographs of the snakes involved, but many of the practitioners lack the expertise to identify the snakes and the potential medical consequences of misidentification can be significant. The authors were interested in whether or not crowdsourcing the identification through web sites might be a viable way to get fast and accurate identifications of the snakes in the photographs. Such identifications might help the physicians choose appropriate treatments with greater accuracy and speed than might be possible without an identification of the snake involved.

The analyses seem reasonable, although I'm not an expert with these statistical methods. It would be helpful for many readers to explain the analyses in more detail, such as stating the dependent and independent variables, covariates, etc. for each analysis.

The key results were that a large number of individuals were available around the clock to provide fast and reasonably accurate identifications of snakes in photographs, but relatively few individuals could identify snakes to the species level.

I found these results interesting, but have difficulty with several aspect of the framework (introduction and discussion) of healthcare applications for this study. I think that both the introduction and discussion would benefit from additional development of key ideas and revision for brevity and conciseness elsewhere.

This study points out some potential implications for snakebite treatment of rapid and accurate identification of the snake involved. However, the use of crowdsourced identifications also has potentially serious legal ramifications that would need to be addressed for before physicians can rely on a crowdsourced identification system in treating snakebites. Such ramifications apply to both healthcare workers as well as the identifiers. A physician relying on crowdsourced identifications is akin to the physician seeking external consultations. The physician would be reluctant to seek input from consultants unless they know the consultant is qualified to provide the information needed, and even then the physician may have to follow norms, guidelines, or perhaps even laws about seeking or using such input. An emergency room physician I've spoken with, who has both experience treating snakebites and extensive knowledge of snakes, felt that a system of training and qualification/certification would be necessary before physicians would be willing to rely on external consultations about snake identification in making decisions about medical treatment.

Lines 245-247 also relate to the point about the qualifications of consultants. The text says: "Many people with expertise in snake identification express that they cannot easily describe how they know the identity of a certain species, particularly in images of poor quality, but that they simply 'just know'." A fundamental problem of crowdsourcing information is the widely variable accuracy and reliability of participants and the information that they provide. How can medical professionals tell when snake identifiers think they "just know" but actually don't know very

well? Crowdsourcing can be self-correcting, but the fundamental difficulty of assessing accuracy remains and may be most critical with potentially life-threatening medical decisions. I think that additional discussion about

On Lines 319-321, the authors note that taxonomic stability is useful to some degree for clinicians, and that the rates at which taxonomic changes are adopted by scientists and non-scientists are not known. I think that some additional points could be discussed here. For example, in some cases, stability of old names for medically important venomous snakes could be helpful to clinicians, as the authors note, but it's also possible that recent taxonomic changes could benefit clinicians by separating populations of venomous snakes whose venom types or toxicities differ. Snake enthusiasts who lack scientific training or understanding may not know about important taxonomic changes, or potentially worse, may choose to disregard them. For example, in an imaginary but possible example, if a widespread viper gets split into multiple species that differ genetically and vary in venom type or toxicity, but otherwise are identical in appearance, then citizen-science identifiers may not know about or may choose to disregard the current names of the snakes, and thus could give wrong and potentially harmful information on the snake in a snakebite case.

Although it was not a goal of this study to develop a crowdsourced identification system for use by physicians, I think it's important at least to acknowledge that developing such a system would involve important ethical and legal considerations as well as medical ones. The manuscript touches on these ideas only very indirectly by mentioning clinician training (e.g., Lines 14-15, 342-344, and 365). For example, on Lines 14-15 (using the numbering column closest to the text), the authors say that "innovative citizen science/crowdsourcing approaches can play significant roles in training and building capacity." What kinds of training and capacity, and how exactly would crowdsourcing approaches enhance training and capacity? Additional explanation and development of the discussion points would be valuable in these places and elsewhere in the manuscript.

Lines 139-140: The authors state that "All data collected were anonymized; thus, human subjects institutional review board approval was not required." More detail about this would be helpful. From the Snake ID Challenge web site, it looks like a username and password are required to participate, and an email address is optional. Participants could have chosen to use their real names in their usernames or email addresses. In those cases, anonymizing the data and which data get stored for future analyses become important. So it would be helpful to have brief explanations of what data were collected about the participants, how the data were anonymized, and what data were stored. I noticed that although participant names were not used, some of the comments in the file about image metadata mistakes contains names and initials that could be used to identify individuals. I wonder if similar information could be included in other supplementary material/files. These potential identifiers must be removed. Furthermore, the fact that they are present in stored files means that this work could already require human subjects review and approval.

Lines 17-18 say "we suggest that healthcare workers, clinicians, epidemiologists, and other parties interested in snakebite become involved in these communities more formally and more often." Similarly, Lines 342-344 state: "We encourage healthcare workers, clinicians, and epidemiologists to become involved in online snake enthusiast communities to learn and build their capacity through practice and participation." It seems like an unrealistic idea that healthcare practitioners have enough time or interest to learn about snake identification through participation in online communities. More specific, targeted, and time-efficient activities, especially ones that draw on certified expertise, are more likely to be useful for continuing professional development by healthcare practitioners than vague encouragement to get more involved with snake enthusiasts. Can you envision more specific ways that online communities

can become more accessible, targeted, demonstrated to be reliable, and therefore more useful to healthcare practitioners?

The image recognition capabilities available online now are often highly accurate, and improving rapidly. It seems likely that in the foreseeable future, automated image recognition will become both more accurate and more widely available. With that development, crowdsourcing identification seems likely to become less important, although perhaps still useful as a check on automated image recognition systems. I can imagine an online system that uses automated image recognition combined with crowdsourcing and vetting by trained experts to provide not only fast recognition of snakes in photos, but also statistics on the current accuracy of the recognition systems based on the vetting and information about the expertise behind the vetting. Lines 309-311 touch on these ideas, but would benefit from further development and being moved to other parts of the manuscript. By providing information (snake identifications), a measure of its accuracy, and data on the expertise behind it, such a system might become much more useful to medical professionals than current crowdsourced citizen-science web sites that give no indication of accuracy or the expertise involved in reaching an identification. I think this manuscript should acknowledge the potential utility of automated image recognition, perhaps coupled with the development of additional data that physicians need to make informed decisions about treatment of venomous snakebite. It seems to me that the more foresight and specific ideas provided in this manuscript, the more valuable it will be to the target audience.

Some minor points:

I recommend being careful to explain what specifically is meant by “herpetologist” on Lines 16-17 and 366-367. Many non-professionals, even non-scientists, probably call themselves herpetologists, whereas the term is probably meant to reflect formally trained biologists with college or higher degrees and formal scientific experience.

Lines 18-19 say that “involving snake enthusiasts in diagnosis and documentation could build the capacity of healthcare workers to identify snakes more quickly, specifically, and accurately, and ultimately improve snakebite treatment data and outcomes.” I don’t think that the authors intended for the term “diagnosis” to mean medical diagnosis, but it would be wise to change the term to avoid this implication.

Lines 58-69: I’m not sure what the main point of this paragraph is for this study. The efficacy of treatment with antivenoms is beyond the scope of this study. Perhaps the first point here is that accurate identification of the snakes is important in assessing the efficacy of antivenom treatments for snakebites retrospectively. Although this is true, it is a digression from the focus of this study. I think it would better as a more concise point in the discussion than it is as a long paragraph in the introduction, or possibly just to delete this material. A good secondary point here is clearer: that accurate identification is sometimes necessary to distinguish snakes whose bites may have mild to moderate effects but don’t usually need antivenom treatment from those whose bites are more severe and often require antivenom treatment. This point relates more directly to the framework for this study, but it could be made more concisely.

Lines 104-105: The authors state that “assess the potential role of citizen science and crowdsourcing in supporting snake ecology and snakebite epidemiology and management.” The text about snake ecology should be removed because this manuscript doesn’t address snake ecology.

Lines 292-311: I think that this paragraph can be shortened considerably and still make the key point that crowdsourcing can sometimes provide more accuracy than individuals working alone.

Lines 342-344: In addition to, or perhaps instead of, encouraging medical professionals to learning more about snake identification by becoming involved in online communities of snake enthusiasts, why not encourage knowledgeable identifiers to become involved in targeted MIVS databases/web sites/projects/etc.?

It also would be helpful for readers interested in exploring the data if the supplemental file of responses data file included fields indicating whether answers were correct and incorrect.

Review form: Reviewer 2

Is the manuscript scientifically sound in its present form?

Yes

Are the interpretations and conclusions justified by the results?

No

Is the language acceptable?

No

Do you have any ethical concerns with this paper?

No

Have you any concerns about statistical analyses in this paper?

Yes

Recommendation?

Major revision is needed (please make suggestions in comments)

Comments to the Author(s)

This manuscript is really interesting in that it uses a crowd-sourcing platform to evaluate the accuracy of snake identification by committed members of the public. I have very few major concerns, but they do need some attention or clarification. Most concerns are relatively minor and are enumerated below.

One of the major concerns is that your statistical approach is explained only very briefly (in a single paragraph) and it is not clear exactly what you did or how.

How many modeling approaches do you use?

How many models do you make in each approach?

What are the response variables?

Are you using logistic regression for correct ID or not?

Or are you using proportion correct as your response variable? If so, did you transform the data since proportion will be between 0-1?

Did you check your statistical assumptions? Perform any transformations of the data?

How did you compare the models? You say in lines 147-148 that you tested which variable had the greatest impact on accuracy or speed, but did you do this by comparing models with AIC? Or are you referring to effect size as your comparison? Relative variable importance? Very unclear.

Also, there are statistical packages in R that can handle mixed effects models to allow you to control for participant.

It's difficult to know whether your approach is sound without edits to this section. Please revise to carefully explain what your modeling approach is and how it relates to the hypotheses or predictions of your manuscript.

Minor concerns:

L1) Delete 'do you know these snakes?' from the title. Unnecessary

L51) Citations 3-5 aren't very apt to use here. They don't support the point you are trying to make here, and in fact, this problem occurs several times in the ms.

L77-79 and throughout) why not use the word "avocational herpetologist" here and in other instances? It's probably the most fitting description of many of these participants.

L125) change "scientific binomial" to "scientific binomen" here and throughout.

L163) Some reference to the Pareto principle here may be useful. Your data are a close approximation to the idea that 20% of participants will account for 80% of the data.

L164) by 'these' do you mean "this subset"?

L234-237) this is a very lengthy run-on sentence with troubling verb agreement. Please rewrite to clarify.

L238) change to past tense when describing your results.

L243-244) "is suggestive of the fact"..... "suggests". Don't use more words when a single 1 would suffice.

L278) not sure citation 49 is useful here.

L281) change "more well understood" to "better understood"

L362-366) another lengthy run-on sentence that is difficult to understand here. "together with and mentorship"?

Figure 2A x-axis supposed to say "percent CORRECT" if I understand this right. Also, try open circles and closed circles. It's impossible to distinguish between these triangles and circles.

How about an effect size plot to understand which of the variables have the greatest impact on proportion correctly identified?

Review form: Reviewer 3

Is the manuscript scientifically sound in its present form?

Yes

Are the interpretations and conclusions justified by the results?

Yes

Is the language acceptable?

Yes

Do you have any ethical concerns with this paper?

Yes

Have you any concerns about statistical analyses in this paper?

Yes

Recommendation?

Major revision is needed (please make suggestions in comments)

Comments to the Author(s)

Comments to the Authors

General:

This manuscript describes the results of study of humans' ability to visually identify species and the utility of citizen-science and crowdsourcing approaches to the problem of species identification. The manuscript reads relatively well and does a good job of identifying the problem of achieving accurate snake species identification and its clinical importance. Overall, I would have liked to see this manuscript offer more applications of the methodology/approach and to reach outside of snake species identification. Essentially, this is a manuscript about humans identifying snakes, but this study and its approach could serve as a model for how one might approach the greater species identification problems. The broader impacts then of the study are rather limited, and in a revision the authors should address the other potential impacts (documenting biodiversity, identifying rare and threatened species, etc.). To only make it about snakes and snakebite treatment is very limiting. The readership I believe in general would appreciate broader implications.

I am not sure that this study is in fact exempt from IRB review. It is my understanding that by definition studies involving research with human subjects that include interactions with subjects (even if digital and anonymous) are subject to IRB review. If the IRB of the institution deemed it "Exempt", then this should be stated. If the protocol was not reviewed by IRB, then it should be stated and the publisher should decide whether they want to publish or not based on ethical reasons.

Overall, I would like to have seen a better description or representation of the statistical results. The authors do a good job of stating the results of various comparisons, but are more inconsistent about when and which statistics to report. For instance, in figure S1 there is a graph that shows the "importance" of variables in the model, but there are no actual descriptions of what "importance" means statistically. In addition to the graphs, a summary of the actual model values (r-square, coefficients, p-values) would be very useful.

Smaller detailed revisions by line (I used the original MS line numbers rather than the proof line numbers):

Abstract

Line 14: Insert ", " between the words "engagement" and "and"

Line 20: Delete the ", " between "specifically" and "and"

Introduction

As this is not a taxon-specific journal nor is it a taxon-specific problem, the opening paragraph could start more broadly, as the problem of species identification is not specific to snakes – identification of birds, plants, invertebrates, and basically anything that is identified visually from a limited number of characteristics are all subject to human errors in visual identification and could benefit from the results of this study.

In the introduction overall, I suggest making reference to both these similar issues as well as how they have similarly been addressed so that the authors approach to snakes can be better placed within that context. I believe this can be done without taking away from the broader impacts on snakebite treatment.

Line 29: When using “place to place” are you referring to geographical variation or variation in microhabitat? It may be helpful to clarify.

Line 30: Remove “Even” as it is not needed.

Line 33: Remove “,” after species as it is not needed and makes sentence easier to read.

Line 36: Add “,” between “life-threatening” and “neglected”

Line 73: Provide citation and reference information for FieldHerpForum, HerpMapper.org and iNaturalist as readers unfamiliar with these platforms will not know what they are or how to find them.

Line 74: Facebook should also be cited and referenced

Line 76: Twitter should also be cited and referenced

Methods

I am not sure that this study is in fact exempt from IRB review. It is my understanding that by definition studies involving research with human subjects that include interactions with subjects (even if digital and anonymous) are subject to IRB review. If the IRB of the institution deemed it exempt, then this should be stated. If the protocol was not reviewed by IRB, then it should be stated as such in the methods.

Line 108: Please identify what the PYBOSSA acronym stands for, as readers will not know

Lines 107–134: Here or in a separate paragraph it is important to state here whether the same images or taxa were repeated in a test. For instance, whether a participant may be offered two different images of a *Bitis arietans*. In addition, please provide rationale for why you did or did not repeat images and/or taxa. I gathered from table S4 that repeat images and species were used, but it is a little bit unclear in the text. Please clarify.

Line 135: Provide the date range rather than just the month and the year as this aids in the repeatability of the study

Line 137: Please briefly indicate what a “digital badge” is as readers unfamiliar with this will not know what it refers to. A simple parenthetical description would suffice.

Lines 142–146: Depending on whether you repeated images or taxa for a given test, you should provide whether and how this affected speed, accuracy and precision.

Line 149: Change “better” to “more accurate”

Line 150: Change “they did” to “participants”

Results

I was unable to find most details of the statistical model itself (e.g. coefficients, weights, p-values) in the manuscript or supplements. This information is provided sporadically in the text, but it is both relevant to the results as well as important information for others when analyzing the study or planning future studies. A supplemental table with the model descriptives should be relatively easy to generate and would provide valuable information to readers.

Line 157: Change the hyphen in “1-1,088) to an en dash

Line 161: Add “platforms” directly behind “social media”

Line 163–164: Remove “A subset of” as this makes the sentence confusing.

Line 190: Replace “hard” with “difficult” as this was previously identified in line 116 of methods as terminology used.

Line 191: Remove “marginally” as the next line identifies marginality

Line 192: More accurately identified than snakes from other countries?

Line 203: Please clarify the use of the word “Significantly” here. If it is statistical significance, please provide the statistics. If it is not, please use another word to avoid confusion.

Line 213: Remove the period before “(Fig 2A)”

Line 215: “Significantly better” is used here in a way that is unclear. Is it referring to statistically significant differences? If so, please provide the statistics. And participants are significantly better than what exactly? (better at identifying snakes within versus outside of the participant’s geographic region?)

Discussion

I have mentioned some of the issues in the introduction, and have similar problems with the discussion. In a subsequent revision, some comparisons with studies that have addressed similar problems in other taxa would be beneficial here.

Overall this section reads well and there were only a few additional issues.

The term “hard” is used throughout this section, and while it is assumed that it means the same as “difficult” it is important to clarify whether the participants were given “hard” or “difficult” questions. The authors will need to make sure that all uses of either word match the terms that were used in the game.

Line 328: Replace “hard” with “difficult” as this was previously identified in line 116 of methods as terminology used.

Line 330: Replace “hard” with “difficult” as this was previously identified in line 116 of methods as terminology used.

References

There is inconsistent usage of journal abbreviations throughout this section. Rather than conduct a line-by-line review of this issue here, I recommend that the authors review every reference in this section and utilize appropriate, widely used journal abbreviations for every reference when possible. This is in line with RSOS publishing guidelines.

For all page ranges, an en dash should be used instead of a hyphen. Again, rather a line-by-line review of the issue here, I recommend that the authors review every reference in this section and correct this.

Line 391: I believe the URL needs to be provided for this citation. <http://www.reptile-database.org>. There's also a dangling "(" here. Possibly from code.

Line 464: The full URL should be provided and There's also a dangling "(" at the end.

Line 469: Add "R package version" immediately before "6.0-82" and remove the "("

Figures and Tables

Figure 1:

This figure is missing letters on the actual image (A and B).

I recommend adding "Easy" and "Difficult" to the caption as these are terms that you specifically used in your game in the methods.

I recommend pointing out in the image what it is about each that make it "Easy" or "Difficult" referring to the terms you mention in the methods section.

Figure 2:

The text on these figures needs to be larger.

If possible, the size of the symbols should also be larger, but without having them overlap.

While colors are useful, some readers are colorblind. Variation in shading could be used as a way to fix this problem, but it is up to the authors and publisher if it is required.

Figure 3.

See the color issue mentioned in Figure 2. Again, this is more of the author and publisher preferences and shouldn't prohibit publication.

Table 1.

No comments.

Table 2.

No comments.

Table 3.

No Comments

Supplemental Figures

Figure S1:

The text on this figure is too small.

The caption and Y-axis labels are uninterpretable to anyone who didn't run the model, and either need to be made readable or described in the caption.

The authors need to identify what is meant (statistically) by "importance". I don't believe this is identified anywhere in the manuscript, and would be useful in interpreting results.

A table of the model parameters, coefficients, etc. would also be useful here.

Figure S2:

The text on this figure is better than in others, but is still a little too small.

The symbols and error bars in this figure are too small.

The use of the term "hard" is not consistent with the term "difficult" which is what was stated in the methods as the term used in the game. Please clarify this here and throughout the manuscript.

Decision letter (RSOS-191835.R0)

20-Jan-2020

Dear Mr Durso,

Manuscript ID RSOS-191835 entitled "Do you know these snakes? Crowdsourcing snake identification with online communities of herpetological enthusiasts" which you submitted to Royal Society Open Science, has been reviewed. The comments from reviewers are included at the bottom of this letter.

In view of the criticisms of the reviewers, the manuscript has been rejected in its current form. However, a new manuscript may be submitted which takes into consideration these comments.

Please note that resubmitting your manuscript does not guarantee eventual acceptance, and that your resubmission will be subject to peer review before a decision is made.

Your resubmitted manuscript should be submitted by 19-Jul-2020. If you are unable to submit by this date please contact the Editorial Office.

Best regards,
Lianne Parkhouse
Editorial Coordinator

on behalf of Dr Jake Socha (Associate Editor) and Professor Kevin Padian (Subject Editor)
 openscience@royalsociety.org

Associate Editor Comments to Author (Dr Jake Socha):

The three reviewers were generally enthusiastic about the use of crowdsourcing and online tools as a means of quick species identification. However, there were numerous concerns about the methods and writing that need to be addressed. More importantly, two of the reviewers brought up concerns about the alleged lack of need of IRB approval for a study involving human subjects. Additionally, there are concerns about the re-use of images. These issues are addressed in detail by the Dr. Kevin Padian, the Subject Editor.

Subject Editor Comments to Author (Professor Kevin Padian):

Thanks for your efforts on this very interesting manuscript. The editorial staff discussed the IRB issue with our journal editor and he agrees that, as this research used and studied human responses and interactions with a citizen science project, IRB approval should be obtained. As with many other citizen science projects, there is typically a "User Agreement" that is clearly linked with the log-in/registration page which informs the users how their data will be used (such as with the citizen science projects at www.zooniverse.org), but this appears to be absent from the registration page of the Snake ID Challenge website.

Our editor also noted the use of snake photographs that have been sourced from Twitter. The Twitter terms of use state that re-use is allowed only through their official embed tool; all other instances of re-use require permission from the Twitter users themselves:

"If you want to reproduce, modify, create derivative works, distribute, sell, transfer, publicly display, publicly perform, transmit, or otherwise use the Services or Content on the Services, you must use the interfaces and instructions we provide."

It doesn't appear that the authors have used the official embed tool, which infringes upon Twitter's terms of use. Some of these images of snakes appear to be in the public domain already, but many images are sourced from individual Twitter users, where permission should have been received, OR the Twitter embed tool is used -- <https://developer.twitter.com/en/docs/twitter-for-websites/embedded-tweets/overview>

The other issue is the names within the metadata and the direct URLs to specific Facebook groups (in which user profiles, full names, and responses are clearly visible). This looks like an oversight on the authors' part, so we want to clarify that we can only consider a revised paper if the metadata are completely anonymised.

We will entertain a resubmission, provided that the authors:

- Obtain IRB approval for the study and supply proof of this (for example, a certificate as a file uploaded to their submission, or a signed letter from an ethical approval body);
- Address the potential copyright infringement of the snake photographs sourced from Twitter;
- Anonymise all metadata uploaded to figshare and their supplementary material;
- Provide a thorough response to the additional reviewer concerns.

Thanks for your submission and please respond carefully to all concerns if you choose to resubmit.

Reviewers' Comments to Author:

Reviewer: 1

Comments to the Author(s)

This study used a citizen science web site project with photographs of snakes to assess the ability, availability, speed, and accuracy of participants from around the world to identify snakes, particularly medically important venomous snakes; it also assessed the effects of photograph quality. The authors explain that healthcare practitioners treating snakebite patients are often presented with photographs of the snakes involved, but many of the practitioners lack the expertise to identify the snakes and the potential medical consequences of misidentification can be significant. The authors were interested in whether or not crowdsourcing the identification through web sites might be a viable way to get fast and accurate identifications of the snakes in the photographs. Such identifications might help the physicians choose appropriate treatments with greater accuracy and speed than might be possible without an identification of the snake involved.

The analyses seem reasonable, although I'm not an expert with these statistical methods. It would be helpful for many readers to explain the analyses in more detail, such as stating the dependent and independent variables, covariates, etc. for each analysis.

The key results were that a large number of individuals were available around the clock to provide fast and reasonably accurate identifications of snakes in photographs, but relatively few individuals could identify snakes to the species level.

I found these results interesting, but have difficulty with several aspect of the framework (introduction and discussion) of healthcare applications for this study. I think that both the introduction and discussion would benefit from additional development of key ideas and revision for brevity and conciseness elsewhere.

This study points out some potential implications for snakebite treatment of rapid and accurate identification of the snake involved. However, the use of crowdsourced identifications also has potentially serious legal ramifications that would need to be addressed for before physicians can rely on a crowdsourced identification system in treating snakebites. Such ramifications apply to both healthcare workers as well as the identifiers. A physician relying on crowdsourced identifications is akin to the physician seeking external consultations. The physician would be reluctant to seek input from consultants unless they know the consultant is qualified to provide the information needed, and even then the physician may have to follow norms, guidelines, or perhaps even laws about seeking or using such input. An emergency room physician I've spoken with, who has both experience treating snakebites and extensive knowledge of snakes, felt that a system of training and qualification/certification would be necessary before physicians would be willing to rely on external consultations about snake identification in making decisions about medical treatment.

Lines 245-247 also relate to the point about the qualifications of consultants. The text says: "Many people with expertise in snake identification express that they cannot easily describe how they know the identity of a certain species, particularly in images of poor quality, but that they simply 'just know'." A fundamental problem of crowdsourcing information is the widely variable accuracy and reliability of participants and the information that they provide. How can medical

professionals tell when snake identifiers think they “just know” but actually don’t know very well? Crowdsourcing can be self-correcting, but the fundamental difficulty of assessing accuracy remains and may be most critical with potentially life-threatening medical decisions. I think that additional discussion about

On Lines 319-321, the authors note that taxonomic stability is useful to some degree for clinicians, and that the rates at which taxonomic changes are adopted by scientists and non-scientists are not known. I think that some additional points could be discussed here. For example, in some cases, stability of old names for medically important venomous snakes could be helpful to clinicians, as the authors note, but it’s also possible that recent taxonomic changes could benefit clinicians by separating populations of venomous snakes whose venom types or toxicities differ. Snake enthusiasts who lack scientific training or understanding may not know about important taxonomic changes, or potentially worse, may choose to disregard them. For example, in an imaginary but possible example, if a widespread viper gets split into multiple species that differ genetically and vary in venom type or toxicity, but otherwise are identical in appearance, then citizen-science identifiers may not know about or may choose to disregard the current names of the snakes, and thus could give wrong and potentially harmful information on the snake in a snakebite case.

Although it was not a goal of this study to develop a crowdsourced identification system for use by physicians, I think it’s important at least to acknowledge that developing such a system would involve important ethical and legal considerations as well as medical ones. The manuscript touches on these ideas only very indirectly by mentioning clinician training (e.g., Lines 14-15, 342-344, and 365). For example, on Lines 14-15 (using the numbering column closest to the text), the authors say that “innovative citizen science/crowdsourcing approaches can play significant roles in training and building capacity.” What kinds of training and capacity, and how exactly would crowdsourcing approaches enhance training and capacity? Additional explanation and development of the discussion points would be valuable in these places and elsewhere in the manuscript.

Lines 139-140: The authors state that “All data collected were anonymized; thus, human subjects institutional review board approval was not required.” More detail about this would be helpful. From the Snake ID Challenge web site, it looks like a username and password are required to participate, and an email address is optional. Participants could have chosen to use their real names in their usernames or email addresses. In those cases, anonymizing the data and which data get stored for future analyses become important. So it would be helpful to have brief explanations of what data were collected about the participants, how the data were anonymized, and what data were stored. I noticed that although participant names were not used, some of the comments in the file about image metadata mistakes contains names and initials that could be used to identify individuals. I wonder if similar information could be included in other supplementary material/files. These potential identifiers must be removed. Furthermore, the fact that they are present in stored files means that this work could already require human subjects review and approval.

Lines 17-18 say “we suggest that healthcare workers, clinicians, epidemiologists, and other parties interested in snakebite become involved in these communities more formally and more often.” Similarly, Lines 342-344 state: “We encourage healthcare workers, clinicians, and epidemiologists to become involved in online snake enthusiast communities to learn and build their capacity through practice and participation.” It seems like an unrealistic idea that healthcare practitioners have enough time or interest to learn about snake identification through participation in online communities. More specific, targeted, and time-efficient activities, especially ones that draw on certified expertise, are more likely to be useful for continuing professional development by healthcare practitioners than vague encouragement to get more

involved with snake enthusiasts. Can you envision more specific ways that online communities can become more accessible, targeted, demonstrated to be reliable, and therefore more useful to healthcare practitioners?

The image recognition capabilities available online now are often highly accurate, and improving rapidly. It seems likely that in the foreseeable future, automated image recognition will become both more accurate and more widely available. With that development, crowdsourcing identification seems likely to become less important, although perhaps still useful as a check on automated image recognition systems. I can imagine an online system that uses automated image recognition combined with crowdsourcing and vetting by trained experts to provide not only fast recognition of snakes in photos, but also statistics on the current accuracy of the recognition systems based on the vetting and information about the expertise behind the vetting. Lines 309-311 touch on these ideas, but would benefit from further development and being moved to other parts of the manuscript. By providing information (snake identifications), a measure of its accuracy, and data on the expertise behind it, such a system might become much more useful to medical professionals than current crowdsourced citizen-science web sites that give no indication of accuracy or the expertise involved in reaching an identification. I think this manuscript should acknowledge the potential utility of automated image recognition, perhaps coupled with the development of additional data that physicians need to make informed decisions about treatment of venomous snakebite. It seems to me that the more foresight and specific ideas provided in this manuscript, the more valuable it will be to the target audience.

Some minor points:

I recommend being careful to explain what specifically is meant by “herpetologist” on Lines 16-17 and 366-367. Many non-professionals, even non-scientists, probably call themselves herpetologists, whereas the term is probably meant to reflect formally trained biologists with college or higher degrees and formal scientific experience.

Lines 18-19 say that “involving snake enthusiasts in diagnosis and documentation could build the capacity of healthcare workers to identify snakes more quickly, specifically, and accurately, and ultimately improve snakebite treatment data and outcomes.” I don’t think that the authors intended for the term “diagnosis” to mean medical diagnosis, but it would be wise to change the term to avoid this implication.

Lines 58-69: I’m not sure what the main point of this paragraph is for this study. The efficacy of treatment with antivenoms is beyond the scope of this study. Perhaps the first point here is that accurate identification of the snakes is important in assessing the efficacy of antivenom treatments for snakebites retrospectively. Although this is true, it is a digression from the focus of this study. I think it would better as a more concise point in the discussion than it is as a long paragraph in the introduction, or possibly just to delete this material. A good secondary point here is clearer: that accurate identification is sometimes necessary to distinguish snakes whose bites may have mild to moderate effects but don’t usually need antivenom treatment from those whose bites are more severe and often require antivenom treatment. This point relates more directly to the framework for this study, but it could be made more concisely.

Lines 104-105: The authors state that “assess the potential role of citizen science and crowdsourcing in supporting snake ecology and snakebite epidemiology and management.” The text about snake ecology should be removed because this manuscript doesn’t address snake ecology.

Lines 292-311: I think that this paragraph can be shortened considerably and still make the key point that crowdsourcing can sometimes provide more accuracy than individuals working alone.

Lines 342-344: In addition to, or perhaps instead of, encouraging medical professionals to learning more about snake identification by becoming involved in online communities of snake enthusiasts, why not encourage knowledgeable identifiers to become involved in targeted MIVS databases/web sites/projects/etc.?

It also would be helpful for readers interested in exploring the data if the supplemental file of responses data file included fields indicating whether answers were correct and incorrect.

Reviewer: 2

Comments to the Author(s)

This manuscript is really interesting in that it uses a crowd-sourcing platform to evaluate the accuracy of snake identification by committed members of the public. I have very few major concerns, but they do need some attention or clarification. Most concerns are relatively minor and are enumerated below.

One of the major concerns is that your statistical approach is explained only very briefly (in a single paragraph) and it is not clear exactly what you did or how.

How many modeling approaches do you use?

How many models do you make in each approach?

What are the response variables?

Are you using logistic regression for correct ID or not?

Or are you using proportion correct as your response variable? If so, did you transform the data since proportion will be between 0-1?

Did you check your statistical assumptions? Perform any transformations of the data?

How did you compare the models? You say in lines 147-148 that you tested which variable had the greatest impact on accuracy or speed, but did you do this by comparing models with AIC? Or are you referring to effect size as your comparison? Relative variable importance? Very unclear. Also, there are statistical packages in R that can handle mixed effects models to allow you to control for participant.

It's difficult to know whether your approach is sound without edits to this section. Please revise to carefully explain what your modeling approach is and how it relates to the hypotheses or predictions of your manuscript.

Minor concerns:

L1) Delete 'do you know these snakes?' from the title. Unnecessary

L51) Citations 3-5 aren't very apt to use here. They don't support the point you are trying to make here, and in fact, this problem occurs several times in the ms.

L77-79 and throughout) why not use the word "avocational herpetologist" here and in other instances? It's probably the most fitting description of many of these participants.

L125) change "scientific binomial" to "scientific binomen" here and throughout.

L163) Some reference to the Pareto principle here may be useful. Your data are a close approximation to the idea that 20% of participants will account for 80% of the data.

L164) by 'these' do you mean "this subset"?

L234-237) this is a very lengthy run-on sentence with troubling verb agreement. Please rewrite to clarify.

L238) change to past tense when describing your results.

L243-244) "is suggestive of the fact"..... "suggests". Don't use more words when a single 1 would suffice.

L278) not sure citation 49 is useful here.

L281) change "more well understood" to "better understood"

L362-366) another lengthy run-on sentence that is difficult to understand here. “together with and mentorship”?

Figure 2A x-axis supposed to say “percent CORRECT” if I understand this right. Also, try open circles and closed circles. It’s impossible to distinguish between these triangles and circles.

How about an effect size plot to understand which of the variables have the greatest impact on proportion correctly identified?

Reviewer: 3

Comments to the Author(s)

General:

This manuscript describes the results of study of humans’ ability to visually identify species and the utility of citizen-science and crowdsourcing approaches to the problem of species identification. The manuscript reads relatively well and does a good job of identifying the problem of achieving accurate snake species identification and its clinical importance. Overall, I would have liked to see this manuscript offer more applications of the methodology/approach and to reach outside of snake species identification. Essentially, this is a manuscript about humans identifying snakes, but this study and its approach could serve as a model for how one might approach the greater species identification problems. The broader impacts then of the study are rather limited, and in a revision the authors should address the other potential impacts (documenting biodiversity, identifying rare and threatened species, etc.). To only make it about snakes and snakebite treatment is very limiting. The readership I believe in general would appreciate broader implications.

I am not sure that this study is in fact exempt from IRB review. It is my understanding that by definition studies involving research with human subjects that include interactions with subjects (even if digital and anonymous) are subject to IRB review. If the IRB of the institution deemed it “Exempt”, then this should be stated. If the protocol was not reviewed by IRB, then it should be stated and the publisher should decide whether they want to publish or not based on ethical reasons.

Overall, I would like to have seen a better description or representation of the statistical results. The authors do a good job of stating the results of various comparisons, but are more inconsistent about when and which statistics to report. For instance, in figure S1 there is a graph that shows the “importance” of variables in the model, but there are no actual descriptions of what “importance” means statistically. In addition to the graphs, a summary of the actual model values (r-square, coefficients, p-values) would be very useful.

Smaller detailed revisions by line (I used the original MS line numbers rather than the proof line numbers):

Abstract

Line 14: Insert “,” between the words “engagement” and “and”

Line 20: Delete the “,” between “specifically” and “and”

Introduction

As this is not a taxon-specific journal nor is it a taxon-specific problem, the opening paragraph

could start more broadly, as the problem of species identification is not specific to snakes – identification of birds, plants, invertebrates, and basically anything that is identified visually from a limited number of characteristics are all subject to human errors in visual identification and could benefit from the results of this study.

In the introduction overall, I suggest making reference to both these similar issues as well as how they have similarly been addressed so that the authors approach to snakes can be better placed within that context. I believe this can be done without taking away from the broader impacts on snakebite treatment.

Line 29: When using “place to place” are you referring to geographical variation or variation in microhabitat? It may be helpful to clarify.

Line 30: Remove “Even” as it is not needed.

Line 33: Remove “,” after species as it is not needed and makes sentence easier to read.

Line 36: Add “,” between “life-threatening” and “neglected”

Line 73: Provide citation and reference information for FieldHerpForum, HerpMapper.org and iNaturalist as readers unfamiliar with these platforms will not know what they are or how to find them.

Line 74: Facebook should also be cited and referenced

Line 76: Twitter should also be cited and referenced

Methods

I am not sure that this study is in fact exempt from IRB review. It is my understanding that by definition studies involving research with human subjects that include interactions with subjects (even if digital and anonymous) are subject to IRB review. If the IRB of the institution deemed it exempt, then this should be stated. If the protocol was not reviewed by IRB, then it should be stated as such in the methods.

Line 108: Please identify what the PYBOSSA acronym stands for, as readers will not know

Lines 107–134: Here or in a separate paragraph it is important to state here whether the same images or taxa were repeated in a test. For instance, whether a participant may be offered two different images of a *Bitis arietans*. In addition, please provide rationale for why you did or did not repeat images and/or taxa. I gathered from table S4 that repeat images and species were used, but it is a little bit unclear in the text. Please clarify.

Line 135: Provide the date range rather than just the month and the year as this aids in the repeatability of the study

Line 137: Please briefly indicate what a “digital badge” is as readers unfamiliar with this will not know what it refers to. A simple parenthetical description would suffice.

Lines 142–146: Depending on whether you repeated images or taxa for a given test, you should provide whether and how this affected speed, accuracy and precision.

Line 149: Change “better” to “more accurate”

Line 150: Change “they did” to “participants”

Results

I was unable to find most details of the statistical model itself (e.g. coefficients, weights, p-values) in the manuscript or supplements. This information is provided sporadically in the text, but it is both relevant to the results as well as important information for others when analyzing the study or planning future studies. A supplemental table with the model descriptives should be relatively easy to generate and would provide valuable information to readers.

Line 157: Change the hyphen in “1-1,088) to an en dash

Line 161: Add “platforms” directly behind “social media”

Line 163–164: Remove “A subset of” as this makes the sentence confusing.

Line 190: Replace “hard” with “difficult” as this was previously identified in line 116 of methods as terminology used.

Line 191: Remove “marginally” as the next line identifies marginality

Line 192: More accurately identified than snakes from other countries?

Line 203: Please clarify the use of the word “Significantly” here. If it is statistical significance, please provide the statistics. If it is not, please use another word to avoid confusion.

Line 213: Remove the period before “(Fig 2A)”

Line 215: “Significantly better” is used here in a way that is unclear. Is it referring to statistically significant differences? If so, please provide the statistics. And participants are significantly better than what exactly? (better at identifying snakes within versus outside of the participant’s geographic region?)

Discussion

I have mentioned some of the issues in the introduction, and have similar problems with the discussion. In a subsequent revision, some comparisons with studies that have addressed similar problems in other taxa would be beneficial here.

Overall this section reads well and there were only a few additional issues.

The term “hard” is used throughout this section, and while it is assumed that it means the same as “difficult” it is important to clarify whether the participants were given “hard” or “difficult” questions. The authors will need to make sure that all uses of either word match the terms that were used in the game.

Line 328: Replace “hard” with “difficult” as this was previously identified in line 116 of methods as terminology used.

Line 330: Replace “hard” with “difficult” as this was previously identified in line 116 of methods as terminology used.

References

There is inconsistent usage of journal abbreviations throughout this section. Rather than conduct a line-by-line review of this issue here, I recommend that the authors review every reference in this section and utilize appropriate, widely used journal abbreviations for every reference when possible. This is in line with RSOS publishing guidelines.

For all page ranges, an en dash should be used instead of a hyphen. Again, rather a line-by-line review of the issue here, I recommend that the authors review every reference in this section and correct this.

Line 391: I believe the URL needs to be provided for this citation. <http://www.reptile-database.org>. There's also a dangling "(" here. Possibly from code.

Line 464: The full URL should be provided and There's also a dangling "(" at the end.

Line 469: Add "R package version" immediately before "6.0-82" and remove the "("

Figures and Tables

Figure 1:

This figure is missing letters on the actual image (A and B).

I recommend adding "Easy" and "Difficult" to the caption as these are terms that you specifically used in your game in the methods.

I recommend pointing out in the image what it is about each that make it "Easy" or "Difficult" referring to the terms you mention in the methods section.

Figure 2:

The text on these figures needs to be larger.

If possible, the size of the symbols should also be larger, but without having them overlap.

While colors are useful, some readers are colorblind. Variation in shading could be used as a way to fix this problem, but it is up to the authors and publisher if it is required.

Figure 3.

See the color issue mentioned in Figure 2. Again, this is more of the author and publisher preferences and shouldn't prohibit publication.

Table 1.

No comments.

Table 2.

No comments.

Table 3.

No Comments

Supplemental Figures

Figure S1:

The text on this figure is too small.

The caption and Y-axis labels are uninterpretable to anyone who didn't run the model, and either need to be made readable or described in the caption.

The authors need to identify what is meant (statistically) by "importance". I don't believe this is identified anywhere in the manuscript, and would be useful in interpreting results.

A table of the model parameters, coefficients, etc. would also be useful here.

Figure S2:

The text on this figure is better than in others, but is still a little too small.

The symbols and error bars in this figure are too small.

The use of the term "hard" is not consistent with the term "difficult" which is what was stated in the methods as the term used in the game. Please clarify this here and throughout the manuscript.

Author's Response to Decision Letter for (RSOS-191835.R0)

See Appendix A.

RSOS-201273.R0

Review form: Reviewer 1

Is the manuscript scientifically sound in its present form?

Yes

Are the interpretations and conclusions justified by the results?

Yes

Is the language acceptable?

Yes

Do you have any ethical concerns with this paper?

No

Have you any concerns about statistical analyses in this paper?

No

Recommendation?

Accept with minor revision (please list in comments)

Comments to the Author(s)

This study showed that a large number of individuals are available around the clock to provide fast and reasonably accurate identifications of snakes in photographs, but relatively few individuals could identify snakes to the species level. These identifications have potential value to medical professionals treating patients with snakebite. I found these results interesting, and think the authors have a pretty good job of revising the manuscript in response to the previous reviews.

It would have been helpful if the replies to the reviews pointed to specific page and line numbers where revisions, at least major ones, were made in response to the reviews. The authors did this to a limited degree, but it was frustrating and time-consuming to figure it out on my own when the authors didn't do so. That extra work made me more critical and slower in finishing the review. I strongly recommend pointing to locations of revisions more thoroughly in future response letters. Pointing to specific locations of revisions would make it easier and faster for reviewers to assess the revisions. For example, it was difficult to determine how much of the four paragraphs in the response letter beginning with "The use of crowdsourcing is growing rapidly in healthcare and more widely in global health (Wazny 2018)" were incorporated into the manuscript. The answer was not very much, although at least the most important parts were mentioned briefly in the manuscript. I had similar experiences with some other response comments.

Although it wasn't a goal of the study to develop a method of promoting physician involvement in citizen-science identifications of snakes, it's one thing to recommend such involvement and another thing entirely to accomplish it. I appreciate the revisions that acknowledged the ethical and legal considerations of physicians crowdsourcing snake identifications. However, I'm still skeptical that simply encouraging medical professionals to "become involved in these communities more formally and more often" is meaningful by itself. Most medical professionals are highly unlikely to read this study and are too busy to do this, as the authors know. For these reasons, I think that it would be more meaningful and make this study more useful to a broader audience if the authors could suggest some ways in which the audience of this study, not just medical professionals, could take steps to promote interactions between the two communities. To me it seems more promising to call on herpetologists to organize ways to help connect medical professionals to citizen-science resources. Even still, most herpetologists would not know how to do that. Giving specific suggestions in the manuscript for ways to accomplish this would make the study more broadly useful. For example, how did the authors, and how could others, come to collaborate with the University Hospitals of Geneva, Doctors Without Borders, and the World Health Organisation, and how did they come to give presentations to medical colleagues and students? Similarly, how could one become involved with IUCN Species Specialist Groups, the African Snakebite Institute, First Aid for Snakebite, and similar organizations? Specific suggestions for how to become a 'human in the loop' (as the authors mention on Line 455) could be very helpful to readers who want to follow this study's recommendations for getting involved, but don't know how to start. I think a few minor additions along these lines could make this study more broadly useful, and could help at least some people take steps toward the kinds of engagement that the authors are encouraging.

Decision letter (RSOS-201273.R0)

Dear Mr Durso

On behalf of the Editors, we are pleased to inform you that your Manuscript RSOS-201273 "Crowdsourcing snake identification with online communities of professional herpetologists and avocational snake enthusiasts" has been accepted for publication in Royal Society Open Science subject to minor revision in accordance with the referees' reports. Please find the referees' comments along with any feedback from the Editors below my signature.

Please submit your revised manuscript and required files (see below) no later than 7 days from today's (ie 16-Nov-2020) date. Note: the ScholarOne system will 'lock' if submission of the revision is attempted 7 or more days after the deadline. If you do not think you will be able to meet this deadline please contact the editorial office immediately.

on behalf of Dr Jake Socha (Associate Editor) and Kevin Padian (Subject Editor)
openscience@royalsociety.org

Associate Editor Comments to Author (Dr Jake Socha):

Congratulations on acceptance of this manuscript, which may lead to practical innovations in areas such as dealing with snakebite. In your final submission, please consider revising the text in regard to Reviewer 1's thoughtful comments; such advice has the potential for greatly increasing the impact of the paper beyond academics.

Reviewer comments to Author:

Reviewer: 1

Comments to the Author(s)

This study showed that a large number of individuals are available around the clock to provide fast and reasonably accurate identifications of snakes in photographs, but relatively few individuals could identify snakes to the species level. These identifications have potential value to medical professionals treating patients with snakebite. I found these results interesting, and think the authors have a pretty good job of revising the manuscript in response to the previous reviews.

It would have been helpful if the replies to the reviews pointed to specific page and line numbers where revisions, at least major ones, were made in response to the reviews. The authors did this to a limited degree, but it was frustrating and time-consuming to figure it out on my own when the authors didn't do so. That extra work made me more critical and slower in finishing the review. I strongly recommend pointing to locations of revisions more thoroughly in future response letters. Pointing to specific locations of revisions would make it easier and faster for reviewers to assess the revisions. For example, it was difficult to determine how much of the four paragraphs in the response letter beginning with "The use of crowdsourcing is growing rapidly in healthcare and more widely in global health (Wazny 2018)" were incorporated into the manuscript. The answer was not very much, although at least the most important parts were mentioned briefly in the manuscript. I had similar experiences with some other response comments.

Although it wasn't a goal of the study to develop a method of promoting physician involvement in citizen-science identifications of snakes, it's one thing to recommend such involvement and another thing entirely to accomplish it. I appreciate the revisions that acknowledged the ethical and legal considerations of physicians crowdsourcing snake identifications. However, I'm still skeptical that simply encouraging medical professionals to "become involved in these communities more formally and more often" is meaningful by itself. Most medical professionals are highly unlikely to read this study and are too busy to do this, as the authors know. For these reasons, I think that it would be more meaningful and make this study more useful to a broader audience if the authors could suggest some ways in which the audience of this study, not just medical professionals, could take steps to promote interactions between the two communities. To me it seems more promising to call on herpetologists to organize ways to help connect medical professionals to citizen-science resources. Even still, most herpetologists would not know how to do that. Giving specific suggestions in the manuscript for ways to accomplish this would make the study more broadly useful. For example, how did the authors, and how could others, come to collaborate with the University Hospitals of Geneva, Doctors Without Borders, and the World Health Organisation, and how did they come to give presentations to medical colleagues and students? Similarly, how could one become involved with IUCN Species Specialist Groups, the African Snakebite Institute, First Aid for Snakebite, and similar organizations? Specific suggestions for how to become a 'human in the loop' (as the authors mention on Line 455) could be very helpful to readers who want to follow this study's recommendations for getting involved, but don't know how to start. I think a few minor additions along these lines could make this study more broadly useful, and could help at least some people take steps toward the kinds of engagement that the authors are encouraging.

===PREPARING YOUR MANUSCRIPT===

===PREPARING YOUR REVISION IN SCHOLARONE===

- Any electronic supplementary material (ESM).
- If you are requesting a discretionary waiver for the article processing charge, the waiver form must be included at this step.
- If you are providing image files for potential cover images, please upload these at this step, and inform the editorial office you have done so. You must hold the copyright to any image provided.
- A copy of your point-by-point response to referees and Editors. This will expedite the preparation of your proof.

- Ensure that your data access statement meets the requirements at <https://royalsociety.org/journals/authors/author-guidelines/#data>. You should ensure that you cite the dataset in your reference list. If you have deposited data etc in the Dryad repository, please only include the 'For publication' link at this stage. You should remove the 'For review' link.
- If you are requesting an article processing charge waiver, you must select the relevant waiver option (if requesting a discretionary waiver, the form should have been uploaded at Step 3 'File upload' above).
- If you have uploaded ESM files, please ensure you follow the guidance at <https://royalsociety.org/journals/authors/author-guidelines/#supplementary-material> to include a suitable title and informative caption. An example of appropriate titling and captioning may be found at https://figshare.com/articles/Table_S2_from_Is_there_a_trade-off_between_peak_performance_and_performance_breadth_across_temperatures_for_aerobic_scope_in_teleost_fishes_/3843624.

Author's Response to Decision Letter for (RSOS-201273.R0)

See Appendix B.

Decision letter (RSOS-201273.R1)

Dear Mr Durso,

It is a pleasure to accept your manuscript entitled "Crowdsourcing snake identification with online communities of professional herpetologists and avocational snake enthusiasts" in its current form for publication in Royal Society Open Science.

You can expect to receive a proof of your article in the near future. Please contact the editorial office (openscience_proofs@royalsociety.org) and the production office (openscience@royalsociety.org) to let us know if you are likely to be away from e-mail contact -- if

you are going to be away, please nominate a co-author (if available) to manage the proofing process, and ensure they are copied into your email to the journal.

on behalf of Dr Jake Socha (Associate Editor) and Kevin Padian (Subject Editor)
openscience@royalsociety.org

Appendix A

20-Jan-2020

Dear Mr Durso,

Manuscript ID RSOS-191835 entitled "Do you know these snakes? Crowdsourcing snake identification with online communities of herpetological enthusiasts" which you submitted to Royal Society Open Science, has been reviewed. The comments from reviewers are included at the bottom of this letter.

In view of the criticisms of the reviewers, the manuscript has been rejected in its current form. However, a new manuscript may be submitted which takes into consideration these comments.

Please note that resubmitting your manuscript does not guarantee eventual acceptance, and that your resubmission will be subject to peer review before a decision is made.

Your resubmitted manuscript should be submitted by 19-Jul-2020. If you are unable to submit by this date please contact the Editorial Office.

Best regards,

on behalf of Dr Jake Socha (Associate Editor) and Professor Kevin Padian (Subject Editor)
openscience@royalsociety.org

Associate Editor Comments to Author (Dr Jake Socha):

The three reviewers were generally enthusiastic about the use of crowdsourcing and online tools as a means of quick species identification. However, there were numerous concerns about the methods and writing that need to be addressed. More importantly, two of the reviewers brought up concerns about the alleged lack of need of IRB approval for a study involving human

subjects. Additionally, there are concerns about the re-use of images. These issues are addressed in detail by the Dr. Kevin Padian, the Subject Editor.

Subject Editor Comments to Author (Professor Kevin Padian):

Thanks for your efforts on this very interesting manuscript. The editorial staff discussed the IRB issue with our journal editor and he agrees that, as this research used and studied human responses and interactions with a citizen science project, IRB approval should be obtained. As with many other citizen science projects, there is typically a “User Agreement” that is clearly linked with the log-in/registration page which informs the users how their data will be used (such as with the citizen science projects at www.zooniverse.org), but this appears to be absent from the registration page of the Snake ID Challenge website.

The Swiss Human Research Act (HRA) regulates research involving human beings in Switzerland (see <https://www.admin.ch/opc/en/classified-compilation/20061313/index.html>). We submitted our project on “Crowdsourcing snake identification for biodiversity and global health” to the Cantonal Ethics Committee Geneva (CCER), which confirmed that the aim of this project is outside of the scope of the Swiss law (see letter from “Commission cantonale d'éthique de la recherche (CCER) » Reference: Req-2020-00172). This letter has been uploaded with the resubmitted manuscript. On lines 165-170, we have now replaced “All data collected were anonymized; thus, human subjects institutional review board approval was not required“ with “This study was exempted from Ethics approval by the Cantonal Ethics Committee Geneva because it does not fall within the scope of the Swiss human research act (HRA). All data were anonymised”. We also confirm that the Citizen Science Center Zurich has Terms of Use & Privacy Policy that apply to all users visiting and/or using any of the CS Zurich platforms, including the “Snake identification challenge” (see: <https://citizenscience.ch/en/terms>). These Terms of Use & Privacy Policy are linked at the bottom of all pages of the “Snake identification challenge”.

Our editor also noted the use of snake photographs that have been sourced from Twitter. The Twitter terms of use state that re-use is allowed only through their official embed tool; all other instances of re-use require permission from the Twitter users themselves:

“If you want to reproduce, modify, create derivative works, distribute, sell, transfer, publicly display, publicly perform, transmit, or otherwise use the Services or Content on the Services, you must use the interfaces and instructions we provide.”

It doesn't appear that the authors have used the official embed tool, which infringes upon Twitter's terms of use. Some of these images of snakes appear to be in the public domain already, but many images are sourced from individual Twitter users, where permission should have been received, OR the Twitter embed tool is used --

<https://developer.twitter.com/en/docs/twitter-for-websites/embedded-tweets/overview>

In fact the use of snake photographs that have been sourced from Twitter refers to a different but related project, from which we do not present data here. We apologize for the oversight and will take these comments into account as we prepare that manuscript. The manuscript under review here does not use any images collected from Twitter.

The other issue is the names within the metadata and the direct URLs to specific Facebook groups (in which user profiles, full names, and responses are clearly visible). This looks like an oversight on the authors' part, so we want to clarify that we can only consider a revised paper if the metadata are completely anonymised.

Names appear in the file 'snapp-image-metadata-mistakes.csv' in two columns: 'photographer' and 'notes'. It is easy enough to remove these and we have prepared an anonymised file. We understand the need to protect the personal identification information of the photographers and the Facebook users, but we also wish to 1) respect photographer intellectual property by properly crediting their works and 2) provide transparent information about the source of the images by linking to the original sources. Additionally, some of the photographers are from Europe (GDPR) or California (CCPA), which have stricter attribution requirement statutes than elsewhere. Do you have any suggestions for how we can best do this?

We will entertain a resubmission, provided that the authors:

- Obtain IRB approval for the study and supply proof of this (for example, a certificate as a file uploaded to their submission, or a signed letter from an ethical approval body);
- Address the potential copyright infringement of the snake photographs sourced from Twitter;
- Anonymise all metadata uploaded to figshare and their supplementary material;
- Provide a thorough response to the additional reviewer concerns.

Thanks for your submission and please respond carefully to all concerns if you choose to resubmit.

Reviewers' Comments to Author:

Reviewer: 1

Comments to the Author(s)

This study used a citizen science web site project with photographs of snakes to assess the ability, availability, speed, and accuracy of participants from around the world to identify snakes, particularly medically important venomous snakes; it also assessed the effects of photograph quality. The authors explain that healthcare practitioners treating snakebite patients are often presented with photographs of the snakes involved, but many of the practitioners lack the expertise to identify the snakes and the potential medical consequences of misidentification can be significant. The authors were interested in whether or not crowdsourcing the identification

through web sites might be a viable way to get fast and accurate identifications of the snakes in the photographs. Such identifications might help the physicians choose appropriate treatments with greater accuracy and speed than might be possible without an identification of the snake involved.

The analyses seem reasonable, although I'm not an expert with these statistical methods. It would be helpful for many readers to explain the analyses in more detail, such as stating the dependent and independent variables, covariates, etc. for each analysis.

We now use an entirely new model, in which dependent and independent variables for each analysis are now more clearly stated. In addition, a new co-author (Andrew Kleinhesselink) has been brought on because of his role in improving the data analysis.

The key results were that a large number of individuals were available around the clock to provide fast and reasonably accurate identifications of snakes in photographs, but relatively few individuals could identify snakes to the species level.

I found these results interesting, but have difficulty with several aspect of the framework (introduction and discussion) of healthcare applications for this study. I think that both the introduction and discussion would benefit from additional development of key ideas and revision for brevity and conciseness elsewhere.

This study points out some potential implications for snakebite treatment of rapid and accurate identification of the snake involved. However, the use of crowdsourced identifications also has potentially serious legal ramifications that would need to be addressed for before physicians can rely on a crowdsourced identification system in treating snakebites. Such ramifications apply to both healthcare workers as well as the identifiers. A physician relying on crowdsourced identifications is akin to the physician seeking external consultations. The physician would be reluctant to seek input from consultants unless they know the consultant is qualified to provide the information needed, and even then the physician may have to follow norms, guidelines, or perhaps even laws about seeking or using such input. An emergency room physician I've spoken with, who has both experience treating snakebites and extensive knowledge of snakes, felt that a system of training and qualification/certification would be necessary before physicians would be willing to rely on external consultations about snake identification in making decisions about medical treatment.

The use of crowdsourcing is growing rapidly in healthcare and more widely in global health (Wazny 2018). There are an increasing number of crowdsourcing online platforms supporting large groups of both medical professionals (e.g. CrowdMed, HealthTap) and patients around the world (e.g. PatientsLikeMe), particularly around the diagnosis of complex health conditions. In fact, crowdsourcing diagnosis has attracted the interest of high-impact media and was recently covered by The New York Times and in the documentary Diagnosis on Netflix. This movement should not be ignored and we need to think innovatively about possible applications of this collaborative approach in contexts where the number of trained health professionals is more

limited and where populations affected by complex and traumatising diseases are severely neglected. This is the case of snakebite and other neglected tropical diseases, and certain forms of crowdsourcing are already being used in different parts of the world. Yet, as highlighted by R1, crowdsourcing diagnosis involves certain changes in the way we practice medicine and raises important legal and ethical risks that must be carefully addressed (Sims et al. 2018).

By responding to the zoological question, what is the taxonomic identity of a given snake?, we aim, first of all, to improve the understanding of snake identification in general and more specifically snakebite epidemiology in regions of the world where snakebite is endemic. Shockingly little is known about the identity of snakes that bite people in developing countries. Crowdsourcing snake identification could also improve the clinical management of snakebite, yet this is not our primary objective and it may be limited to certain regions where there is a high risk of snake misidentifications (e.g. high snake diversity, mimics) and of negative consequences of these misidentifications for victims' outcome.

Crowdsourcing snake identification, at least at a small scale, is already used among healthcare providers managing snakebite in the field and illustrates the need for support from the herpetological community. For example, we are aware of the existence of multiple WhatsApp groups or have even identified cases via Facebook where healthcare providers share photos of biting snakes with their medical colleagues and herpetologists they may know and trust, but increasingly beyond their immediate networks. This process has emerged spontaneously and it often occurs in an informal way, with healthcare providers being ultimately responsible for any clinical decision. Today, we have the potential to increase the impact of such a process by tapping on specialised online zoological platforms (e.g. iNaturalist, HerpMapper) or even groups on Facebook and therefore on a much larger and international community of herpetologists, both professionals and hobbyists. Indeed, the speed and accuracy at identifying snakes from photos of this often heterogeneous communities needs to be assessed, which was the primary objective of the submitted study, and factored in in any decision support system.

In the mid-term, our objective would be to develop a snake photo identification system (e.g. mobile app) based on crowdsourcing and an algorithm that generates a possible consensual output by considering all responses from the community and weighing in the level of expertise of each contributor (e.g. based on snake identification challenges as the one submitted here, based on an initial test on snake identification and a questionnaire on the level expertise) (which is in line with the suggestion by R1). The level of certainty (or error) of the output will also be provided and, as any clinical decision support tool, the final decision and the legal responsibility will be in the hands of the healthcare provider, not in the community of herpetologists. Yet, as suggested by R1, a relation of trust between medical professionals and the community of herpetologists needs to be gradually built and collaborations between these two communities, still very limited, need to be promoted. The publication of initiatives like the one proposed here can contribute to this and push for innovative solutions in the context of the WHO's roadmap.

Lines 245-247 also relate to the point about the qualifications of consultants. The text says: "Many people with expertise in snake identification express that they cannot easily describe how they know the identity of a certain species, particularly in images of poor quality, but that they

simply 'just know'." A fundamental problem of crowdsourcing information is the widely variable accuracy and reliability of participants and the information that they provide. How can medical professionals tell when snake identifiers think they "just know" but actually don't know very well? Crowdsourcing can be self-correcting, but the fundamental difficulty of assessing accuracy remains and may be most critical with potentially life-threatening medical decisions. I think that additional discussion about

Somewhat addressed above and in the last paragraph of the intro and 3rd paragraph of the discussion. In the 5th paragraph of the discussion, we also address that identifiers in our challenge could not interact with or correct each other, and we suggest that asking a crowd of people who can interact is most likely to provide a correct result, presumably because more knowledgeable crowd members will guide others to the correct identification (which is what happens in existing online communities of snake enthusiasts). We also suggest combining crowdsourcing with artificial intelligence, which has the potential to produce results superior to those of either one alone (see below).

On Lines 319-321, the authors note that taxonomic stability is useful to some degree for clinicians, and that the rates at which taxonomic changes are adopted by scientists and non-scientists are not known. I think that some additional points could be discussed here. For example, in some cases, stability of old names for medically important venomous snakes could be helpful to clinicians, as the authors note, but it's also possible that recent taxonomic changes could benefit clinicians by separating populations of venomous snakes whose venom types or toxicities differ. Snake enthusiasts who lack scientific training or understanding may not know about important taxonomic changes, or potentially worse, may choose to disregard them. For example, in an imaginary but possible example, if a widespread viper gets split into multiple species that differ genetically and vary in venom type or toxicity, but otherwise are identical in appearance, then citizen-science identifiers may not know about or may choose to disregard the current names of the snakes, and thus could give wrong and potentially harmful information on the snake in a snakebite case.

This is a good point and we agree that your imaginary example is indeed possible. This is one area where a frequently-updated global database of snake range maps combined with an algorithm for narrowing the choices based on the geographic origin of the request would help guide both crowd members and clinicians to the correct name (a good example of this is the filters used to guide eBird reviewers and users to which bird species are expected at a particular date and location). Practically speaking, however, 1) research on venom lags behind splits and other new species descriptions, 2) development of new antivenom lags even farther behind venom research, and 3) many good species already possess venom that varies geographically or ontogenetically (e.g. Rokyta et al. 2013). Also, this problem is already widespread; in our review of 150 publications that contained data on biting snake identification, a substantial number of snakes—1,859 (21%) individuals of 22 species—were reported in the literature using only the common name, and of those reported using the scientific name, 20 species (92 individuals) had been moved to another genus since the time of publication, 10 species (241 individuals) had experienced other taxonomic changes (splits or lumps) (e.g. *Echis ocellatus* in

Cameroon has been renamed *E. romani*), and the names of 3 species (6 individuals) contained minor misspellings as reported (Bolon et al. 2020). Thus, although ignorance of or disregard for the latest taxonomic changes on the part of citizen-science identifiers or clinicians already exists and makes articulating the correct name with particular cases challenging, we doubt that increased interaction between clinicians and citizen-scientists will increase this problem significantly. We now discuss this in more detail in the 6th paragraph of the discussion on lines 466-494.

Although it was not a goal of this study to develop a crowdsourced identification system for use by physicians, I think it's important at least to acknowledge that developing such a system would involve important ethical and legal considerations as well as medical ones. The manuscript touches on these ideas only very indirectly by mentioning clinician training (e.g., Lines 14-15, 342-344, and 365). For example, on Lines 14-15 (using the numbering column closest to the text), the authors say that "innovative citizen science/crowdsourcing approaches can play significant roles in training and building capacity." What kinds of training and capacity, and how exactly would crowdsourcing approaches enhance training and capacity? Additional explanation and development of the discussion points would be valuable in these places and elsewhere in the manuscript.

Somewhat addressed above and in the last paragraph of the intro and 3rd and last paragraphs of the discussion. The capacity in question is to accurately and quickly identify snakes. The training could be as informal as clinicians observing the ID process, thereby becoming familiar with commonly-confused species in their area and the features used to distinguish them, but also asking and receiving answers to questions, including common pitfalls or areas of scientific uncertainty, or as formal as classes led by experts. Existing community group administrators also curate open collections of reference materials that are available to all group members (including any clinicians who join the groups) as public files, containing links to frequently asked questions, diagrams showing salient morphological features, range maps, and other such resources. Finally, repeated exposure to images of relevant snake species (or to living and dead individuals of the snakes themselves) is probably the best way to become familiar with them.

Lines 139-140: The authors state that "All data collected were anonymized; thus, human subjects institutional review board approval was not required." More detail about this would be helpful. From the Snake ID Challenge web site, it looks like a username and password are required to participate, and an email address is optional. Participants could have chosen to use their real names in their usernames or email addresses. In those cases, anonymizing the data and which data get stored for future analyses become important. So it would be helpful to have brief explanations of what data were collected about the participants, how the data were anonymized, and what data were stored. I noticed that although participant names were not used, some of the comments in the file about image metadata mistakes contains names and initials that could be used to identify individuals. I wonder if similar information could be included in other supplementary material/files. These potential identifiers must be removed. Furthermore,

the fact that they are present in stored files means that this work could already require human subjects review and approval.

How the data were anonymized is now detailed in the 2nd paragraph of the methods. We have double-checked all supplementary material and removed any potential identifiers (see above). Names appeared only in the file 'snapp-image-metadata-mistakes.csv' in two columns: 'photographer' and 'notes'. It is easy enough to remove these and I have prepared an anonymised file. We understand the need to protect the privacy of the photographers and the Facebook users, but we also wish to 1) respect photographer intellectual property by properly crediting their works and 2) provide transparent information about the source of the images by linking to the original sources. Do you have any suggestions for how we can best do this?

Lines 17-18 say “we suggest that healthcare workers, clinicians, epidemiologists, and other parties interested in snakebite become involved in these communities more formally and more often.” Similarly, Lines 342-344 state: “We encourage healthcare workers, clinicians, and epidemiologists to become involved in online snake enthusiast communities to learn and build their capacity through practice and participation.” It seems like an unrealistic idea that healthcare practitioners have enough time or interest to learn about snake identification through participation in online communities. More specific, targeted, and time-efficient activities, especially ones that draw on certified expertise, are more likely to be useful for continuing professional development by healthcare practitioners than vague encouragement to get more involved with snake enthusiasts. Can you envision more specific ways that online communities can become more accessible, targeted, demonstrated to be reliable, and therefore more useful to healthcare practitioners?

We agree with R1 that healthcare providers are overloaded and that extra educational and/or training activities often need to be certified and carefully integrated into already heavy curricula and clinical activities. We cannot expect healthcare providers to also become snake biologists and the relevance of their knowledge in snake identification will depend on the endemicity of snakebite and the diversity of snakes in the region where they work. There are certain certified courses and workshops on snakebite management targeting healthcare providers in snakebite endemic regions and the number and quality of these courses is expected to increase in the coming years with the roll out of WHO's snakebite roadmap in each country.

Our online platform and the associated snake identification exercise that we describe in this manuscript could be integrated into these courses and workshops to develop local and national snake identification skills in an innovative and gamified way. We already collaborate with the University Hospitals of Geneva, Doctors Without Borders, the World Health Organisation and have multiple connections in snakebite endemic countries (e.g. Nepal, Cameroun, India) to explore and test this novel training approach. More generally, we believe that younger healthcare providers and medical students, particularly those interested in Tropical Medicine and snakebite, should be aware of well organised online biodiversity platforms such as INaturalist and HerpMapper and of the relevance of such vibrant and international communities

of snake biologists. Interestingly, every time we have presented these platforms to medical colleagues and students, they have generated great curiosity and interest among them and some have now become users. Further dissemination efforts are required across both the medical and the herpetological communities.

Finally, several important groups in snake identification training and snakebite management include avocational members, including some IUCN Species Specialist Groups and the African Snakebite Institute, a private company which is the leading training provider of Snake Awareness, First Aid for Snakebite, and Venomous Snake Handling courses in Africa and the largest distributor of quality snake handling equipment on the continent, with a disproportionate impact on non-healthcare snake ID training.

We have revised our language on lines 547-557 in an attempt to better reflect that we did not mean that these communities should replace certified identification programs, and elaborate on how our data can be used to certify participants in online communities, with an eye towards creating a hybrid community that leverages the size of online communities and the rigor of certified programs (see lines 450-456).

The image recognition capabilities available online now are often highly accurate, and improving rapidly. It seems likely that in the foreseeable future, automated image recognition will become both more accurate and more widely available. With that development, crowdsourcing identification seems likely to become less important, although perhaps still useful as a check on automated image recognition systems. I can imagine an online system that uses automated image recognition combined with crowdsourcing and vetting by trained experts to provide not only fast recognition of snakes in photos, but also statistics on the current accuracy of the recognition systems based on the vetting and information about the expertise behind the vetting. Lines 309-311 touch on these ideas, but would benefit from further development and being moved to other parts of the manuscript. By providing information (snake identifications), a measure of its accuracy, and data on the expertise behind it, such a system might become much more useful to medical professionals than current crowdsourced citizen-science web sites that give no indication of accuracy or the expertise involved in reaching an identification. I think this manuscript should acknowledge the potential utility of automated image recognition, perhaps coupled with the development of additional data that physicians need to make informed decisions about treatment of venomous snakebite. It seems to me that the more foresight and specific ideas provided in this manuscript, the more valuable it will be to the target audience.

We agree about the foresight and specific ideas, and we now elaborate on this more on lines 450-456. In fact what you suggest (“an online system that uses automated image recognition combined with crowdsourcing and vetting by trained experts to provide not only fast recognition of snakes in photos, but also statistics on the current accuracy of the recognition systems based on the vetting and information about the expertise behind the vetting”) is exactly what we are in the process of building. We are preparing another manuscript describing our automated image recognition system, in which we discuss many of these ideas and directly compare the AI performance on the same images with that of participants in this challenge

(spoiler: accuracy and error rates are currently similar, and depend on the species of snake). We agree that image recognition capabilities available online are improving rapidly, but our research on this topic has revealed that crowd labeling of data for AI training is an essential component of creating a functional AI. Neural networks are very hungry for labeled data, and expert labeling is arduous and inefficient.

We have spent the last 2 years amassing the largest collection of images of snakes ever (almost half a million images, in collaboration with almost every sizable online data source [iNaturalist, HerpMapper, Flickr, Twitter, and Facebook], to be described in a 3rd paper) which are currently in their 4th round of AI training/algorithm development here. Despite the promise of this dataset, many species are still underrepresented and we have more research to do to understand where AI and humans can complement each other and how many images are needed per species, and only crowds of people can collect and contribute new images. We have strengthened some of these points in the manuscript, but we hope you will understand that we are preparing other manuscripts that focus more strongly on the interplay among AI, crowds, and experts, and we lack sufficient space here to go into great detail about these issues. This paper is intended to be a building block for those.

Some minor points:

I recommend being careful to explain what specifically is meant by “herpetologist” on Lines 16-17 and 366-367. Many non-professionals, even non-scientists, probably call themselves herpetologists, whereas the term is probably meant to reflect formally trained biologists with college or higher degrees and formal scientific experience.

Agreed, we have revised the manuscript to use “herpetologist” only in the context of a formally trained biologist, and “avocational snake enthusiast” for people who know a lot about snake identification but are not formally trained in it.

Lines 18-19 say that “involving snake enthusiasts in diagnosis and documentation could build the capacity of healthcare workers to identify snakes more quickly, specifically, and accurately, and ultimately improve snakebite treatment data and outcomes.” I don’t think that the authors intended for the term “diagnosis” to mean medical diagnosis, but it would be wise to change the term to avoid this implication.

We now use the term “decision-making” although our original meaning was in fact medical diagnosis (i.e. the identification of a snake, which would narrow a diagnosis from “snakebite” to e.g. “*Echis* bite” or “puff adder bite”).

Lines 58-69: I’m not sure what the main point of this paragraph is for this study. The efficacy of treatment with antivenoms is beyond the scope of this study. Perhaps the first point here is that accurate identification of the snakes is important in assessing the efficacy of antivenom treatments for snakebites retrospectively. Although this is true, it is a digression from the focus of this study. I think it would better as a more concise point in the discussion than it is as a long

paragraph in the introduction, or possibly just to delete this material. A good secondary point here is clearer: that accurate identification is sometimes necessary to distinguish snakes whose bites may have mild to moderate effects but don't usually need antivenom treatment from those whose bites are more severe and often require antivenom treatment. This point relates more directly to the framework for this study, but it could be made more concisely.

Our intent was to emphasize that the need for snake identification is twofold: it's sometimes useful to involve an expert in the clinical treatment, but consultation with an expert is useful even after the patient has left the hospital, because retrospective snake identification is still useful for epidemiology and antivenom efficacy studies. In fact, given the legal and ethical challenges, this will probably be the primary role for crowdsourcing in snakebite research. We don't view this as a digression because it is important for readers to understand that bites from different snake species require different antivenoms. Trust me, it is shocking how many health professionals are not familiar with this.

Lines 104-105: The authors state that "assess the potential role of citizen science and crowdsourcing in supporting snake ecology and snakebite epidemiology and management." The text about snake ecology should be removed because this manuscript doesn't address snake ecology.

We agree and have removed "snake ecology" here

Lines 292-311: I think that this paragraph can be shortened considerably and still make the key point that crowdsourcing can sometimes provide more accuracy than individuals working alone.

We agree and have shortened this paragraph

Lines 342-344: In addition to, or perhaps instead of, encouraging medical professionals to learning more about snake identification by becoming involved in online communities of snake enthusiasts, why not encourage knowledgeable identifiers to become involved in targeted MIVS databases/web sites/projects/etc.?

If targeted MIVS databases existed, we would do so, but none do to our knowledge. We already highlight the MIVS project on iNaturalist. We also wish to emphasize that some patients are bitten by non-venomous snakes but seek treatment anyway, and it's easy for clinicians to forget that there are so many non-venomous snakes, so it's a good idea for this to go both ways. In our experience, online communities of snake enthusiasts do not need to be reminded that they should welcome newcomers (of whatever background) because they are already approached by novices wanting to learn more on a regular basis. Medical professionals, on the other hand, may feel that they already know what they need to know about snake ID, even if this is not the case. We have revised these lines so that this is clearer.

It also would be helpful for readers interested in exploring the data if the supplemental file of responses data file included fields indicating whether answers were correct and incorrect.

This information is available in the 'score' field of 'snapp_responses.csv' and further calculated in fields generated by the code available on figshare. Because we used a somewhat complex scoring system in order to gamify the challenge, the scores are not straightforward to convert to correct/incorrect because an answer can be correct at e.g. the genus level but incorrect at the species level, and different numbers of points were awarded for easy vs difficult images. We can include all this in the data file if the editor requires it, but we prefer to ask readers to rely on the provided commented code to do this job. Note that code and data for our new analysis is available on github instead of figshare (the figshare data remain available but are no longer referenced in the paper, all data necessary to replicate the analyses are available on github).

Reviewer: 2

Comments to the Author(s)

This manuscript is really interesting in that it uses a crowd-sourcing platform to evaluate the accuracy of snake identification by committed members of the public. I have very few major concerns, but they do need some attention or clarification. Most concerns are relatively minor and are enumerated below.

One of the major concerns is that your statistical approach is explained only very briefly (in a single paragraph) and it is not clear exactly what you did or how.

How many modeling approaches do you use?

We originally used two separate modeling approaches but we have revised the manuscript to unify these into a single, easier-to-understand modeling approach (a Bayesian Item Response Theory model designed for analyzing data from multiple-choice exams).

How many models do you make in each approach?

We created a set of 6 candidate models and present results from the one with the lowest LOOIC score (see Table S4).

What are the response variables? Are you using logistic regression for correct ID or not?

The response variable is always the accuracy of the participant's response, coded as a numeric response variable (0, 1, 2, 3) to represent the ordinal states ("incorrect at the family level" < "correct at the family but not the genus level" < "correct at the genus but not the species level" < "correct at the species level"). Answers given at the genus level only (e.g. "Thamnophis") and those given at the species level but suggesting an incorrect species within a correct genus (e.g.

“Thamnophis radix” when the correct answer was “Thamnophis sirtalis”) were coded the same (both would get a 2); same for family.

The biggest downside to treating ordinal variables as numeric is that we assume that the numerical distance between each set of subsequent categories is equal (e.g. it is “twice as wrong” to get the family correct as it is to get the species correct). However, given that there are no strict criteria for defining taxonomic families or genera (or even species) and that calibrating each distance using time-calibrated phylogenetic trees would be arduous, we feel comfortable with this approximation even though it implies that there is a bit more information contained in our response variable than there really is. The assumption that the distances are approximately equal seems reasonably close to reality to us. The advantages are that we had much greater flexibility in our choice of analysis, we preserve the information in the ordering (unlike e.g. using an unordered factor), and we are able to use a technique that is widely understood/familiar.

Or are you using proportion correct as your response variable? If so, did you transform the data since proportion will be between 0-1?

We’re now using a totally different model (Bayesian IRT). See above for description of response variable.

Did you check your statistical assumptions? Perform any transformations of the data?

We’re now using a totally different model (Bayesian IRT), and quality checks for convergence are detailed in the supplementary methods. Followed advice of Bürkner, P.-C. (2017). *Advanced Bayesian Multilevel Modeling with the R Package brms*. arXiv:1705.11123.

How did you compare the models? You say in lines 147-148 that you tested which variable had the greatest impact on accuracy or speed, but did you do this by comparing models with AIC? Or are you referring to effect size as your comparison? Relative variable importance? Very unclear.

We created a set of 6 candidate models and ranked them using LOOIC (the Bayesian equivalent of AIC). See Table S4.

Also, there are statistical packages in R that can handle mixed effects models to allow you to control for participant.

We simplified our approach and all models now contain random effects for participant, and for individual images nested within snake species. We also compare variance associated with these two effects in order to describe how much each contributed to the difficulty of the task.

It’s difficult to know whether your approach is sound without edits to this section. Please revise to carefully explain what your modeling approach is and how it relates to the hypotheses or predictions of your manuscript.

We now use a completely new model and have tried to do a better job describing the statistical model that was used. We apologize for the lack of clarity in the initial submission and we believe that we have now improved the clarity and quality of the analysis, largely thanks to contributions from new co-author Andrew Kleinhesselink.

Minor concerns:

L1) Delete 'do you know these snakes?' from the title. Unnecessary

Done

L51) Citations 3-5 aren't very apt to use here. They don't support the point you are trying to make here, and in fact, this problem occurs several times in the ms.

This problem was caused by citation management software that failed to update the numeric reference scheme when citation strings of >2 consecutively-numbered references were used. We have corrected the issue in the EndNote template and these reference strings now refer to the correct references. We apologize for the oversight.

L77-79 and throughout) why not use the word "avocational herpetologist" here and in other instances? It's probably the most fitting description of many of these participants.

We now use "(skilled) avocational snake enthusiasts" as a compromise with other reviewers' suggestions that "herpetologist" refers exclusively to someone with professional training (although this may be in the eye of the beholder). In general we feel that this is largely semantic given the excellent performance of some participants without formal training in our challenge, but many people are very particular about or defensive of their titles, so we opt to annoy as few people as possible.

L125) change "scientific binomial" to "scientific binomen" here and throughout.

Done

L163) Some reference to the Pareto principle here may be useful. Your data are a close approximation to the idea that 20% of participants will account for 80% of the data.

After some research into the Pareto principle, with which we were unfamiliar, we have elected not to reference it directly here. There seems to be some reasonable doubt about its utility, with some social scientists going so far as to call it a tautology, frequently misapplied, or unclear in its limits. In addition, this isn't really what our paper is about, and we certainly would not have thought of the Pareto principle if our data (which are closer to a 90%-20% split anyway) had been otherwise.

L164) by 'these' do you mean "this subset"?

Yes, we have revised the wording here

L234-237) this is a very lengthy run-on sentence with troubling verb agreement. Please rewrite to clarify.

We have revised the wording here

L238) change to past tense when describing your results.

Done

L243-244) “is suggestive of the fact”..... “suggests”. Don’t use more words when a single 1 would suffice.

Done

L278) not sure citation 49 is useful here.

We have removed this citation

L281) change “more well understood” to “better understood”

Done

L362-366) another lengthy run-on sentence that is difficult to understand here. “together with and mentorship”?

We have revised the wording here, and apologize for the typo (“with and”).

Figure 2A x-axis supposed to say “percent CORRECT” if I understand this right. Also, try open circles and closed circles. It’s impossible to distinguish between these triangles and circles.

We have revised this figure.

How about an effect size plot to understand which of the variables have the greatest impact on proportion correctly identified?

We have now plotted effect sizes in Figs. 2 and 3.

Reviewer: 3

Comments to the Author(s)

General:

This manuscript describes the results of study of humans' ability to visually identify species and the utility of citizen-science and crowdsourcing approaches to the problem of species identification. The manuscript reads relatively well and does a good job of identifying the problem of achieving accurate snake species identification and its clinical importance. Overall, I would have liked to see this manuscript offer more applications of the methodology/approach and to reach outside of snake species identification. Essentially, this is a manuscript about humans identifying snakes, but this study and its approach could serve as a model for how one might approach the greater species identification problems. The broader impacts then of the study are rather limited, and in a revision the authors should address the other potential impacts (documenting biodiversity, identifying rare and threatened species, etc.). To only make it about snakes and snakebite treatment is very limiting. The readership I believe in general would appreciate broader implications.

We have added some broader impacts

I am not sure that this study is in fact exempt from IRB review. It is my understanding that by definition studies involving research with human subjects that include interactions with subjects (even if digital and anonymous) are subject to IRB review. If the IRB of the institution deemed it "Exempt", then this should be stated. If the protocol was not reviewed by IRB, then it should be stated and the publisher should decide whether they want to publish or not based on ethical reasons.

Please see above

Overall, I would like to have seen a better description or representation of the statistical results. The authors do a good job of stating the results of various comparisons, but are more inconsistent about when and which statistics to report. For instance, in figure S1 there is a graph that shows the "importance" of variables in the model, but there are no actual descriptions of what "importance" means statistically. In addition to the graphs, a summary of the actual model values (r-square, coefficients, p-values) would be very useful.

We have tried to do a better job describing the statistical model that was used. We apologize for the lack of clarity in the initial submission and we believe that we have now improved the clarity and quality of the analysis, largely thanks to contributions from new co-author Andrew Kleinhesselink.

Smaller detailed revisions by line (I used the original MS line numbers rather than the proof line numbers):

Abstract

Line 14: Insert ", " between the words "engagement" and "and"

Done

Line 20: Delete the “,” between “specifically” and “and”

Done

Introduction

As this is not a taxon-specific journal nor is it a taxon-specific problem, the opening paragraph could start more broadly, as the problem of species identification is not specific to snakes—identification of birds, plants, invertebrates, and basically anything that is identified visually from a limited number of characteristics are all subject to human errors in visual identification and could benefit from the results of this study.

The opening paragraph is now broader and snakes do not come up until the 2nd paragraph.

In the introduction overall, I suggest making reference to both these similar issues as well as how they have similarly been addressed so that the authors approach to snakes can be better placed within that context. I believe this can be done without taking away from the broader impacts on snakebite treatment.

We now open the first paragraph more broadly. We also address successes with other taxa and the utility of our results in the 7th paragraph of the introduction.

Line 29: When using “place to place” are you referring to geographical variation or variation in microhabitat? It may be helpful to clarify.

We have replaced this with “geographical variation”

Line 30: Remove “Even” as it is not needed.

Done

Line 33: Remove “,” after species as it is not needed and makes sentence easier to read.

Done

Line 36: Add “,” between “life-threatening” and “neglected”

Done

Line 73: Provide citation and reference information for FieldHerpForum, HerpMapper.org and iNaturalist as readers unfamiliar with these platforms will not know what they are or how to find them.

All of the guidelines I found suggest that when referring to a website in general in the text of a paper, one needs only to give the URL to the homepage in-text; a full reference at the end is not necessary (e.g. here). We have added “.com” and “.org” where it was missing. What would a full reference at the end even contain beyond the url? These sites don't have single authors or editors or DOIs, and guidelines on how to cite them (e.g. here) direct researchers to citing specific downloads or records (which we have done in our data). Even putting a date accessed is absurd, because we access these sites every day. Finally, the suggestion that a reader will not know how to find a website that is the top hit on any search engine doesn't seem to give readers much credit.

Line 74: Facebook should also be cited and referenced

All of the guidelines I found suggest that when referring to a website in general in the text of a paper (especially one as ubiquitous as Facebook), one needs only to give the URL to the homepage in-text; a full reference at the end is not necessary (e.g. here, here). Table S1 also provides direct URLs to snake ID and snakebite support Facebook groups.

Line 76: Twitter should also be cited and referenced

All of the guidelines I found suggest that when referring to a website in general in the text of a paper (especially one as ubiquitous as Facebook), one needs only to give the URL to the homepage in-text; a full reference at the end is not necessary (e.g. here).

Methods

I am not sure that this study is in fact exempt from IRB review. It is my understanding that by definition studies involving research with human subjects that include interactions with subjects (even if digital and anonymous) are subject to IRB review. If the IRB of the institution deemed it exempt, then this should be stated. If the protocol was not reviewed by IRB, then it should be stated as such in the methods.

Please see above.

Line 108: Please identify what the PYBOSSA acronym stands for, as readers will not know

Pybossa is not an acronym, although it is usually written in all caps. We have changed this to “Pybossa” to make it more clear.

Lines 107–134: Here or in a separate paragraph it is important to state here whether the same images or taxa were repeated in a test. For instance, whether a participant may be offered two different images of a *Bitis arietans*. In addition, please provide rationale for why you did or did not repeat images and/or taxa. I gathered from table S4 that repeat images and species were used, but it is a little bit unclear in the text. Please clarify.

This is now more clearly stated. We did not repeat images but we did repeat taxa because we needed replication of the same observer within a taxon.

Line 135: Provide the date range rather than just the month and the year as this aids in the repeatability of the study

Done

Line 137: Please briefly indicate what a “digital badge” is as readers unfamiliar with this will not know what it refers to. A simple parenthetical description would suffice.

Done

Lines 142–146: Depending on whether you repeated images or taxa for a given test, you should provide whether and how this affected speed, accuracy and precision.

We tested whether seeing the same taxa again influenced accuracy and report the results in the last paragraph of the results “Participant learning over time”. Speed did not vary in a meaningful way for any metric we tested. We now explicitly evaluate the “taxa repeat” effect, which we found to have a very minor impact on identification accuracy.

Line 149: Change “better” to “more accurate”

Done

Line 150: Change “they did” to “participants”

Done

Results

I was unable to find most details of the statistical model itself (e.g. coefficients, weights, p-values) in the manuscript or supplements. This information is provided sporadically in the text, but it is both relevant to the results as well as important information for others when analyzing the study or planning future studies. A supplemental table with the model descriptives should be relatively easy to generate and would provide valuable information to readers.

We have tried to do a better job describing the statistical model that was used. We apologize for the lack of clarity in the initial submission and we believe that we have now improved the clarity and quality of the analysis, largely thanks to contributions from new co-author Andrew Kleinhesselink.

Line 157: Change the hyphen in “1-1,088) to an en dash

Done

Line 161: Add “platforms” directly behind “social media”

Done

Line 163–164: Remove “A subset of” as this makes the sentence confusing.

Done

Line 190: Replace “hard” with “difficult” as this was previously identified in line 116 of methods as terminology used.

We have now called “hard”/“difficult” images “low-quality” and “easy” images “high-quality” throughout, as this is what we were really trying to represent, and to avoid confusion with the latent parameter “difficulty” in our new IRT model

Line 191: Remove “marginally” as the next line identifies marginality

Done

Line 192: “North American snakes were” more accurately identified than snakes from other countries?

This sentence no longer appears.

Line 203: Please clarify the use of the word “Significantly” here. If it is statistical significance, please provide the statistics. If it is not, please use another word to avoid confusion.

We have avoided all discussion of statistical significance in alignment with the Bayesian nature of our model.

Line 213: Remove the period before “(Fig 2A)”

Done

Line 215: “Significantly better” is used here in a way that is unclear. Is it referring to statistically significant differences? If so, please provide the statistics. And participants are significantly better than what exactly? (better at identifying snakes within versus outside of the participant’s geographic region?)

We have avoided all discussion of statistical significance in alignment with the Bayesian nature of our model. This should be clearer now.

Discussion

I have mentioned some of the issues in the introduction, and have similar problems with the discussion. In a subsequent revision, some comparisons with studies that have addressed similar problems in other taxa would be beneficial here.

Overall this section reads well and there were only a few additional issues.

The term “hard” is used throughout this section, and while it is assumed that it means the same as “difficult” it is important to clarify whether the participants were given “hard” or “difficult” questions. The authors will need to make sure that all uses of either word match the terms that were used in the game.

We have now called “hard”/“difficult” images “low-quality” and “easy” images “high-quality” throughout, as this is what we were really trying to represent, and to avoid confusion with the latent parameter “difficulty” in our new IRT model

Line 328: Replace “hard” with “difficult” as this was previously identified in line 116 of methods as terminology used.

We have now called “hard”/“difficult” images “low-quality” and “easy” images “high-quality” throughout, as this is what we were really trying to represent, and to avoid confusion with the latent parameter “difficulty” in our new IRT model

Line 330: Replace “hard” with “difficult” as this was previously identified in line 116 of methods as terminology used.

We have now called “hard”/“difficult” images “low-quality” and “easy” images “high-quality” throughout, as this is what we were really trying to represent, and to avoid confusion with the latent parameter “difficulty” in our new IRT model

References

There is inconsistent usage of journal abbreviations throughout this section. Rather than conduct a line-by-line review of this issue here, I recommend that the authors review every reference in this section and utilize appropriate, widely used journal abbreviations for every reference when possible. This is in line with RSOS publishing guidelines.

Done. Both Proceedings B (to which this manuscript was originally submitted) and RSOS offer a flexible format at initial submission, with formatting to follow review and/or acceptance.

For all page ranges, an en dash should be used instead of a hyphen. Again, rather a line-by-line review of the issue here, I recommend that the authors review every reference in this section and correct this.

Done. Both Proceedings B (to which this manuscript was originally submitted) and RSOS offer a flexible format at initial submission, with formatting to follow review and/or acceptance.

Line 391: I believe the URL needs to be provided for this citation. <http://www.reptile-database.org>. There's also a dangling "(" here. Possibly from code.

We have corrected this

Line 464: The full URL should be provided and There's also a dangling "(" at the end.

We have corrected this

Line 469: Add "R package version" immediately before "6.0-82" and remove the "("

We have corrected this

Figures and Tables

Figure 1:

This figure is missing letters on the actual image (A and B).

I recommend adding "Easy" and "Difficult" to the caption as these are terms that you specifically used in your game in the methods.

I recommend pointing out in the image what it is about each that make it "Easy" or "Difficult" referring to the terms you mention in the methods section.

Figure 1 now includes "(left)" and "(right)" and the caption describes the features that we thought made these images low- or high-quality. We have now called "hard"/"difficult" images "low-quality" and "easy" images "high-quality" throughout, as this is what we were really trying to represent, and to avoid confusion with the latent parameter "difficulty" in our new IRT model

Figure 2:

The text on these figures needs to be larger.

If possible, the size of the symbols should also be larger, but without having them overlap.

While colors are useful, some readers are colorblind. Variation in shading could be used as a way to fix this problem, but it is up to the authors and publisher if it is required.

Figure 2 now shows these data in a different format. We have enlarged the text and symbol size. If the publisher requires additional differentiation among series, we can accommodate.

Figure 3.

See the color issue mentioned in Figure 2. Again, this is more of the author and publisher preferences and shouldn't prohibit publication.

Figure 3 now shows these data in a different format. If the publisher requires additional differentiation among series, we can accommodate.

Table 1.

No comments.

Table 2.

No comments.

Table 3.

No Comments

Supplemental Figures

Figure S1:

The text on this figure is too small.

The caption and Y-axis labels are uninterpretable to anyone who didn't run the model, and either need to be made readable or described in the caption.

The authors need to identify what is meant (statistically) by "importance". I don't believe this is identified anywhere in the manuscript, and would be useful in interpreting results.

A table of the model parameters, coefficients, etc. would also be useful here.

Figure S1 now shows other data. A table of the model parameters is now Table 1.

Figure S2:

The text on this figure is better than in others, but is still a little too small.

The symbols and error bars in this figure are too small.

Figure S2 now shows other data, and we have adjusted the text size.

The use of the term “hard” is not consistent with the term “difficult” which is what was stated in the methods as the term used in the game. Please clarify this here and throughout the manuscript.

We have now called “hard”/“difficult” images “low-quality” and “easy” images “high-quality” throughout, as this is what we were really trying to represent, and to avoid confusion with the latent parameter “difficulty” in our new IRT model

Appendix B

Associate Editor Comments to Author (Dr Jake Socha):

Congratulations on acceptance of this manuscript, which may lead to practical innovations in areas such as dealing with snakebite. In your final submission, please consider revising the text in regard to Reviewer 1's thoughtful comments; such advice has the potential for greatly increasing the impact of the paper beyond academics.

Reviewer comments to Author:

Reviewer: 1

Comments to the Author(s)

This study showed that a large number of individuals are available around the clock to provide fast and reasonably accurate identifications of snakes in photographs, but relatively few individuals could identify snakes to the species level. These identifications have potential value to medical professionals treating patients with snakebite. I found these results interesting, and think the authors have a pretty good job of revising the manuscript in response to the previous reviews.

Thank you; we appreciate your time and your thoughtful review!

It would have been helpful if the replies to the reviews pointed to specific page and line numbers where revisions, at least major ones, were made in response to the reviews. The authors did this to a limited degree, but it was frustrating and time-consuming to figure it out on my own when the authors didn't do so. That extra work made me more critical and slower in finishing the review. I strongly recommend pointing to locations of revisions more thoroughly in future response letters. Pointing to specific locations of revisions would make it easier and faster for reviewers to assess the revisions. For example, it was difficult to determine how much of the four paragraphs in the response letter beginning with "The use of crowdsourcing is growing rapidly in healthcare and more widely in global health (Wazny 2018)" were incorporated into the manuscript. The answer was not very much, although at least the most important parts were mentioned briefly in the manuscript. I had similar experiences with some other response comments.

We apologize for the shortcomings of the revision format and we will consider this in future response letters.

Although it wasn't a goal of the study to develop a method of promoting physician involvement in citizen-science identifications of snakes, it's one thing to recommend such involvement and another thing entirely to accomplish it. I appreciate the revisions that acknowledged the ethical and legal considerations of physicians crowdsourcing snake identifications. However, I'm still skeptical that simply encouraging medical professionals to "become involved in these communities more formally and more often" is meaningful by itself. Most medical professionals are highly unlikely to read this study and are too busy to do this, as the authors know. For these reasons, I think that it would be more meaningful and make this study more useful to a broader audience if the authors could suggest some ways in which the audience of this study, not just medical professionals, could take steps to promote interactions between the two communities. To me it seems more promising to call on herpetologists to organize ways to help connect medical professionals to citizen-science resources. Even still, most herpetologists would not know how to do that. Giving specific suggestions in the manuscript for ways to accomplish

this would make the study more broadly useful. For example, how did the authors, and how could others, come to collaborate with the University Hospitals of Geneva, Doctors Without Borders, and the World Health Organisation, and how did they come to give presentations to medical colleagues and students? Similarly, how could one become involved with IUCN Species Specialist Groups, the African Snakebite Institute, First Aid for Snakebite, and similar organizations? Specific suggestions for how to become a 'human in the loop' (as the authors mention on Line 455) could be very helpful to readers who want to follow this study's recommendations for getting involved, but don't know how to start. I think a few minor additions along these lines could make this study more broadly useful, and could help at least some people take steps toward the kinds of engagement that the authors are encouraging.

We appreciate the perspective and have attempted to revise our language accordingly throughout.

Specifically, on line 18: after "we suggest that healthcare workers, clinicians, epidemiologists, and other parties interested in snakebite", we changed "become involved in these communities more formally and more often" to "could become more connected to these communities" and added "and that professional herpetologists and skilled avocational snake enthusiasts could organize ways to help connect medical professionals to crowdsourcing platforms."

In the conclusion section, we highlight that "social-media based 'communities of practice' support dispersed communities and provide forums for learning that are integrated into channels that many people are already using" with the intention of making clear that the barrier to joining one of these communities is low. In the first author's 12 years of experience co-moderating a large snake identification Facebook group (>200,000 members), there are numerous healthcare professionals who are members of these groups and use them informally as a resource, so we don't 100% agree with the reviewer's statement that "medical professionals are too busy to do this".

Our specific recommendation is that informal communities (social media groups and citizen science projects) are likely to provide the greatest gain in access to identification capacity for the lowest investment (essentially nothing, since most people including medical professionals are already social media users), in comparison to taking the time to become involved with formal snake biology or even snakebite-focused associations. We now detail this on lines 522-533 in order to help readers who don't know where to start, and refer to Table S1 there.

In Table S1, we provide a list of specific Facebook groups that are useful communities for anyone to get involved in making these connections—whether a herpetologist or snake enthusiast who wants to contribute their knowledge, or a medical professional who has questions. There are even snakebite support Facebook groups which, although targeted mostly at patients, include many doctors and nurses who offer advice and learn from their colleagues as well as from snake experts. Given that these Facebook groups are public on the web to anyone with a free Facebook account (which many people have already) we suggest that referring readers to Table S1 would be the best way to accomplish the recommendation of the reviewer, with which we agree. In our opinion, these Facebook groups are the most active and useful resources for fast, accurate snake identification, and we wish to make readers aware of them.